# Genetic-optimised aperiodic code for distributed optical fibre sensors

Xizi Sun [1,4], Zhisheng Yang [2,4✉], Xiaobin Hong [1✉], Simon Zaslawski [2], Sheng Wang [1], Marcelo A. Soto [3], Xia Gao [1], Jian Wu [1] & Luc Thévenaz [2]

Distributed optical fibre sensors deliver a map of a physical quantity along an optical fibre, providing a unique solution for health monitoring of targeted structures. Considerable developments over recent years have pushed conventional distributed sensors towards their ultimate performance, while any significant improvement demands a substantial hardware overhead. Here, a technique is proposed, encoding the interrogating light signal by a single-sequence aperiodic code and spatially resolving the fibre information through a fast post-processing. The code sequence is once forever computed by a specifically developed genetic algorithm, enabling a performance enhancement using an unmodified conventional config-uration for the sensor. The proposed approach is experimentally demonstrated in Brillouin and Raman based sensors, both outperforming the state-of-the-art. This methodological breakthrough can be readily implemented in existing instruments by only modifying the software, offering a simple and cost-effective upgrade towards higher performance for dis-tributed fibre sensing.

[1] State Key Laboratory of Information Photonics & Optical Communications, Beijing University of Posts and Telecommunications, Beijing 100876, China. [2] EPFL Swiss Federal Institute of Technology, Institute of Electrical Engineering, SCI STI LT, Station 11, CH-1015 Lausanne, Switzerland. [3] Department of Electronics Engineering, Universidad Técnica Federico Santa María, Valparaíso, Chile. [4] These authors contributed equally: Xizi Sun, Zhisheng Yang. ✉email: zhisheng.yang@epfl.ch; xbhong@bupt.edu.cn

Distributed optical fibre sensors (DOFS)[1] offer the capability to continuously inform on the spatial distribution of environmental quantities (such as temperature/strain[2–16], acoustic impedance[17,18], refractive index[19], etc.) along with an optical fibre. Most DOFS exploit natural scattering processes present in optical fibres[20], such as Rayleigh[2–4,10], spontaneous Raman[5,6,11] and spontaneous/stimulated Brillouin[9,12–19] scatterings. These backscattered optical signals can be interrogated in time[2–8,16–18], frequency[9–11], correlation[12,13,19] or mixed[14,15] domains, each with their proper advantages targeting specific applications[1].

Time-domain approaches based on optical time-domain reflectometry (OTDR)[21] turn out to be intrinsically best suited for long-distance sensing (e.g. tens of km), since the sensing range is limited by the laser coherence length in frequency-domain approaches[1,10], and by background noises originating from uninterrogated fibre positions in correlation-domain approaches[1,12]. Time-domain approaches enable to spatially resolve the local environmental information by analysing the back-reflected response from an intense optical pulse. The pulse energy is positively related to the measurement signal-to-noise ratio (SNR), which in turn ultimately scales and trades-off all critical sensing specifications (i.e. spatial resolution (SR), measurand accuracy, sensing range and measurement time)[1,22–24]. For any conventional time-domain DOFS, the maximum attainable SNR turns out to be limited by fundamental physical constraints: whilst the maximum pulse peak power is limited by the onset of nonlinear effects[25–29], the minimum noise power is determined by the signal bandwidth thus being irreducible whatever the denoising technique used[30,31].

To further improve the optimised performance of conventional time-domain DOFS, the energy of the sensing response must be enhanced without impairing the SR nor activating nonlinear effects. This can be realised by using advanced techniques, such as distributed or lumped optical amplification[32–35], smart signal modulation[4,16,36–39], optical pulse coding[40–61] and any combination of them[62,63]. Among these approaches, optical pulse coding is conceptually the most cost-effective, which operates by launching optical pulse trains into the sensing fibre and demodulating (decoding) the backscattered multi-pulse response through post-processing, delivering an SNR-improved equivalent single-pulse response. Extensive studies have been carried out to maximise the coding operation efficiency[40–64] (i.e. maximising the SNR improvement, so-called coding gain, whilst minimising the hardware overhead and extra measurement time); however, all currently existing code types present fundamental and/or practical limitations. For instance, although codes formed by a single sequence[40,56–62,64] are ideal for minimising measurement time and data storage, they either exhibit an imperfect peak-to-sidelobes ratio after the decoding[40,56,64] or are periodic/cyclic codes[57–62,64] that suffer from impairments induced by signal-dependent noise[1] (resulting in a compromised coding gain that has not yet been quantified in literature). As more robust alternatives, distortion-free aperiodic codes constituted by several sequences, such as Golay[41–47], Simplex codes[48–52,63] and their derivatives[54,55], have been mostly used for DOFS. Both codes can provide SNR enhancement with the same number of total acquisitions as in single-pulse DOFS; however, the additional operation time (e.g. the decoding time and the codeword switching time) and the larger occupied memory (for storing distinct coded traces) compromise the coding operation efficiency. More importantly, to reduce the decoding distortion, additional devices are used to alleviate the decaying envelope of the pulse sequences imposed by the gain saturation of erbium-doped fibre amplifiers (EDFAs)[46,50,53]. Nevertheless, the unevenness of the amplified pulse train cannot be fully suppressed[53], and remaining distortions are still detrimental for certain types of codes (e.g. Golay codes[53]). All above-mentioned issues make the conceptual advantages of the coding technique to be hardly realised in practice to match the vast diversity of real-world applications.

Here, we propose an approach based on the concept of deconvolution, offering the possibility of a single-sequence aperiodic code and overcoming all above-mentioned drawbacks inherent to conventional codes. This enables, for the first time to the best of our knowledge, substantial performance improvement over optimised single-pulse DOFS without any hardware extension. The aperiodic code sequence is once forever optimised by a dedicatedly developed distributed genetic algorithm (DGA), here designated as genetic-optimised code (GO-code), offering a coding gain comparable to that of conventional codes such as Golay and Simplex. More importantly, the flexibility in the code length and the adaptability to the non-uniform amplitude over the pulse sequence (namely the pulse sequence envelop) secure a maximum coding efficiency for any given experimental condition, whilst the minimised operation time and the fast decoding process enables on-line real-time measurements, all being crucially advantageous over conventional codes. The proposed technique can be readily implemented in any classical time-domain DOFS by only modifying the software in relation to the pulse generation and post-processing (decoding), showing an unmatched practical significance. All these advantages are here experimentally validated using the most standard implementations of distributed sensors based on Brillouin optical time-domain analysis (BOTDA) and Raman OTDR (ROTDR). It must be emphasised that all demonstrations for both GO-code and single-pulse schemes are performed under *fully optimised* conditions, i.e. using maximised signal powers[25,26,28] and minimised noise bandwidth (by matching the noise bandwidth to the signal bandwidth through digital filtering[31]), thus securing fair comparisons.

## Results

**Description of the proposed coding technique.** The proposed technique can retrieve the single-pulse response with an improved SNR by only modifying the software of a standard single-pulse distributed sensor in relation to pulse generation (upper row in Fig. 1) and post-processing (bottom row in Fig. 1). In this case, an $N_c$-point discrete-time signal $c(n)$, representing the digitally coded pulse sequence, is uploaded into a programmable pulse generator (e.g. field-programmable gate array, FPGA), which in turn converts $c(n)$ into an electrical signal driving the optical setup. The integer $n$ is linearly related to the time $t$ through the sampling rate $f_s$, so as $t = n/f_s$. The digital coding sequence $c(n)$ is once forever generated from a $N_u$-point binary unipolar sequence $u(n)$ following the three steps described in Fig. 1. Here, $u(n)$ contains 0's and 1's elements and must be dedicatedly designed, as will be elaborated in the next subsection.

Although the amplitude of $c(n)$ is uniform in the digital domain, the coded optical pulse sequence launched into the fibre (blue dots in the middle row of Fig. 1), designated as $c_f(n)$, may exhibit a non-uniform amplitude envelop (grey dashed curve in the middle row of Fig. 1) due to the gain saturation of the EDFA used to boost the optical sequence power. Note that all sequences defined hereafter with a subscript '$f$' are associated with such a non-uniform amplitude envelope. The acquired coded fibre response $r_c^m(n)$ can be commonly expressed by the *linear convolution*[65] between the optical code sequence $c_f(n)$ launched into the fibre and the fibre impulse response $h(n)$, merged with a zero-mean additive noise $e_c(n)$

$$r_c^m(n) = c_f(n) \otimes h(n) + e_c(n). \tag{1}$$

Defining the fibre length $L_h$, the number of samples in the fibre impulse response $h(n)$ is $N_h = 2L_h f_s/v_g$, which further yields the

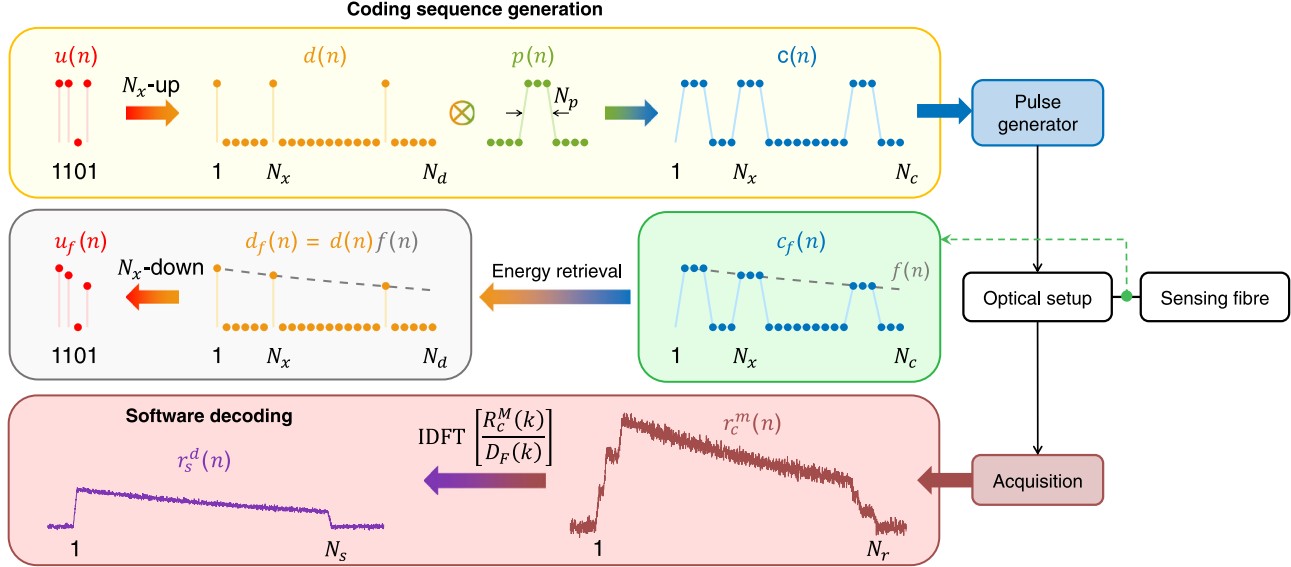

**Fig. 1 Principle of the proposed coding and decoding process.** The upper row shows the procedure to generate an $N_c$-point discrete-time signal $c(n)$ representing the digitally coded pulse sequence from a $N_u$-point unipolar sequence $u(n)$. This procedure involves three steps: (1) generate an $N_p$-point single-pulse signal $p(n)$ depending on the target spatial resolution $\Delta z$, as $N_p = 2\Delta z f_s/v_g$ where $v_g$ is the pulse group velocity in the fibre and $f_s$ is the sampling rate; (2) generate an $N_d$-point sequence $d(n)$ by performing an $N_x$-point upsampling on $u(n)$, so that $N_d = N_x N_u$, where $N_x = N_p$ for NRZ format and $N_x > N_p$ for RZ format. Note that an RZ format must be used in given DOFS with a sufficiently long bit duration ($=N_x/f_s$) to get rid of undesired crosstalk effects that can be imposed by the acoustic inertial response in Brillouin sensing[51] or by the amplified spontaneous forward Raman scattering in ROTDR[57]; and (3) linearly convolve $d(n)$ with $p(n)$, i.e. $c(n) = d(n) \otimes p(n)$, where the sign $\otimes$ denotes linear convolution so that the number of points in the code sequence $c(n)$ is $N_c = N_d + N_p - 1$. The king the energy of each pu middle row shows the actual optical sequence launched into the sensing fibre, denoted as $c_f(n)$, which exhibits an uneven amplitude envelop imposed by a function $f(n)$ determined by the EDFA gain saturating response. By taking the energy of each pulse in $c_f(n)$, an $N_d$-point sequence $d_f(n)$ that is equal to $f(n)d(n)$ can be retrieved and is used for decoding. The bottom row shows the simulated coded fibre response $r_c^m(n)$ and the decoded single-pulse response $r_s^d(n)$.

number of samples in the coded fibre response $r_c^m(n)$ and the noise component $e_c(n)$ to be $N_r = N_c + N_h - 1$. In Eq. (1) the coded optical sequence $c_f(n)$ can be considered as the linear convolution between an $N_d$-point sequence $d_f(n)$ and the single pulse signal $p(n)$, where $d_f(n) = f(n)d(n)$, $f(n)$ being the envelope function of $c_f(n)$ determined by the EDFA gain saturation, as shown by the grey dashed curve in the middle row of Fig. 1. This way, Eq. (1) and its discrete frequency-domain representation can be expressed as

$$r_c^m(n) = d_f(n) \otimes p(n) \otimes h(n) + e_c(n) \xleftrightarrow{F} R_c^M(k) \qquad (2)$$
$$= D_F(k)P(k)H(k) + E_C(k),$$

where $F$ denotes the $N_r$-point discrete Fourier transform (DFT) operation[65].

Here, we designate $D_F(k)$ as the decoding function, which is known since its time-domain representation, $d_f(n)$, can be precisely and easily obtained by retrieving the energy of each optical pulse in the optical code sequence $c_f(n)$ measured at the fibre input (see middle row of Fig. 1). Such calibration can be made for a long measurement session, provided that the experimental stability is sufficient and there is no uncontrolled signal drift of significant importance in the instrumentation. Based on Eq. (2) and inverse DFT[65], the targeted single-pulse response $r_s(n)$ ($=p(n) \otimes h(n)$) can be retrieved by performing the following decoding process:

$$r_s^d(n) = \text{IDFT}\left[\frac{R_c^M(k)}{D_F(k)}\right] = \text{IDFT}\left[P(k)H(k) + \frac{E_C(k)}{D_F(k)}\right] \qquad (3)$$
$$= r_s(n) + \text{IDFT}\left[\frac{E_C(k)}{D_F(k)}\right],$$

where $r_s^d(n)$ is the decoded single-pulse response that contains the

targeted single-pulse response $r_s(n)$ and a noise term affected by the decoding function $D_F(k)$. Equation (3) highlights that the non-distorted single-pulse fibre response $r_s(n)$ can be recovered regardless of the optical coding sequence distortion introduced by the EDFA gain saturation, unlike other conventional codes (e.g. Simplex and Golay) that strictly require a uniform optical code sequence, i.e. $f(n) \equiv 1$. In addition, the decoding process can be made very fast since the DFT can be computed via fast Fourier transform[65].

In Eq. (3), the measure of whether the $D_F(k)$ attenuates or magnifies the original input noise is given by a noise scaling factor $Q$ (for details see Supplementary Eqs. (8)–(12)), which represents the ratio of the noise variance after ($\sigma_{de}^2$, see Supplementary Eq. (8)) and before decoding ($\sigma_e^2$, see Supplementary Eq. (4))

$$Q = \frac{\sigma_{de}^2}{\sigma_e^2} = \frac{1}{N_r}\sum_{k=-\frac{N_r}{2}}^{\frac{N_r}{2}-1}\left|\frac{1}{U_F(k)}\right|^2, \qquad (4)$$

where $U_F(k)$ is the $N_r$-point DFT of the sequence $u_f(n)$ (with an amplitude envelope given by the EDFA gain response), the latter being obtained by an $N_x$-point downsampling of $d_f(n)$, as shown in the middle row of Fig. 1. This way the coding gain (usually defined as the reduction in the noise standard deviation, i.e. $\sigma_e/\sigma_{de}$, like all other codes for DOFS) resulting from our proposed code is $G_c = \sqrt{1/Q}$. It appears evident that to maximise the coding gain, it is required to dedicatedly design and optimise $u_f(n)$.

**Design and optimisation of the code.** The design and optimisation of $u_f(n)$ should firstly maximise the optical energy increment provided by the coded optical pulse sequence $c_f(n)$ with respect to the single optical pulse $p(n)$. Such energy increment is

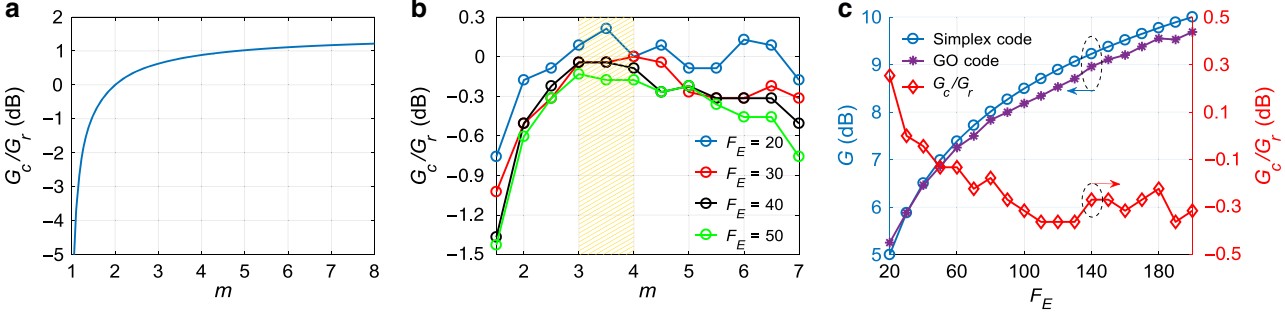

**Fig. 2 Theoretical results of the coding gains in logarithmic scale. a** Theoretical maximum and **b** genetic-optimised $G_c/G_r$ as a function of $m$. **c** Coding gains of the genetic-optimised code and Simplex code (left-hand side vertical axis) and their difference (right-hand side vertical axis) as a function of the energy enhancement factor $F_E$ for $m = 3$.

represented by an *energy enhancement factor* $F_E = \sum_{n=1}^{N_u} u_f(n)$, which is a generic parameter valid for any type of coding. Note that $F_E$ equals to the number of coded pulses only when the code assumes a uniform amplitude envelope (the case of other conventional codes). In general, the upper bound of $F_E$ for any code is ultimately limited by the onset of undesired physical phenomena that get more severe for a larger $F_E$, such as high-order pump depletion[53] and/or additional Brillouin-gain dependent noises (see Supplementary Note 3) in BOTDA, and amplified shot-noise in ROTDR (see Supplementary Note 4). However, for other conventional codes, $F_E$ (the number of coded pulses) is additionally constrained by their mathematical rules[42,48], so that the selected $F_E$ is restricted to specific values that could be much smaller than the ideal upper bound. This limitation makes them less flexible and efficient than the proposed code.

After determining the optimum $F_E$ (the above mentioned upper bound value), the code length $N_u$ should be adapted accordingly (i.e. the larger the $F_E$, the longer the $N_u$). This essentially means optimising the ratio $m = N_u/F_E$ ($m \geq 1$), which is related to the theoretical maximum coding gain $G_c$ (derived based on Eq. (4), see Supplementary Eqs. (12)–(19)):

$$G_c < \sqrt{\frac{F_E(m-1)}{m}}. \tag{5}$$

To evaluate $G_c$, Eq. (5) is compared with *standard reference coding gain* $G_r = \sqrt{F_E/2}$ offered by other conventional unipolar codes commonly used in DOFS, thus further yielding

$$\frac{G_c}{G_r} < \sqrt{\frac{2(m-1)}{m}}. \tag{6}$$

It must be mentioned that not all standard reference coding gain $G_r$ can be practically realised since not every value of $F_E$ is allowed by conventional codes as mentioned before, hence here $G_r$ only represents a numerical reference to evaluate the performance of the proposed technique. Equation (6) suggests that the larger the value of $m$, the larger the upper limit of $G_c/G_r$, as illustrated by Fig. 2a in logarithmic scale. To make $G_c/G_r$ closest possible to this upper limit, the detailed code distribution in $u_f(n)$ must be dedicatedly designed. This is equivalent to design 1's and 0's distribution in $u(n)$, since $u_f(n)$ equals to $u(n)$ being modulated by a known decaying envelope (see Fig. 1).

It turns out that the ideal design should make the off-peak spectral region of $|U_F(k)|^2$ (the power spectrum of the designed code) as flat as possible for given $F_E$ and $m$, as elaborated in Supplementary Note 1, which is extremely complex and cannot be analytically solved to the best of our knowledge. As an alternative, here a DGA[66] is dedicatedly developed to efficiently search for optimal solutions, so that the code proposed here is designated as GO-code. The general concept and detailed

implementation of the DGA are introduced in Methods, and its performance is elaborated in Supplementary Note 2 along with some intermediate results. The optimum value of $m$ for a given energy enhancement factor $F_E$ is empirically found to be between 3 and 4, as illustrated in Fig. 2b. The reason comes from the fact that the high computational complexity in the case of long code lengths (large values of $F_E$ and $m$) leads to a reduction of the genetic optimisation efficiency. Following this criterion, the optimum values of $G_c$ achieved by the DGA searching as a function of $F_E$ are shown in Fig. 2c for $m = 3$ (note that similar curves have also been obtained for $m = 4$, showing only minor differences in terms of gain). It can be found that when $F_E = 20$, the coding gain $G_c$ is 0.3 dB larger than the standard reference coding gain $G_r$; however, the larger the energy enhancement factor $F_E$, the more demanding the computational capability, thus leading to a larger compromise of $G_c$ with respect to $G_r$. Nevertheless, the maximum difference is always <0.5 dB, which can be considered negligible in most applications.

**GO-coded BOTDA.** The performance of the proposed GO-code for BOTDA is investigated and compared with an optimised standard single-pulse BOTDA using the same experimental configuration (see "Methods"). In the setup, the single-pulse and GO-code schemes can be readily switched by alternating the pulse and coding sequence driving the FPGA. For both schemes, the SR is set to 2 m, corresponding to a pulse duration of 20 ns and an optimised single-pulse peak Brillouin gain of 2.5% at the fibre near-end. Other detailed experimental parameters are listed in Table 1.

For the GO-code scheme, the optimum energy enhancement factor $F_E$ is determined to be ~40, to ensure that the coding gain at the fibre far-end is not degraded by additional Brillouin-gain-dependent noises (see details in Supplementary Note 3). Then, following the procedure described in Algorithm 1, and using parameters listed in Table 2, a 135-bit GO-code sequence is found by the proposed DGA. According to Eq. (4), such GO-code

**Table 1 Experimental parameters for both single-pulse and GO-coded schemes.**

**Experimental parameters**

| |
| --- |
| Scanning range 150 MHz |
| Scanning step 1 MHz |
| Frequency switching time $t_s = 5$ ms |
| Acquisition time per trace $t_a = 1.1$ ms |
| Number of trace averages $N_A = 1024$ |
| [a]Measurement time $t_m = 2.8$ min |

[a]$t_m$ = scanning range/scanning step × ($t_a \times N_A + t_s$).

sequence is expected to offer a coding gain of $G_c = 6.2$ dB, being only 0.3 dB lower than the standard reference coding gain ($G_r = 6.5$ dB, $\sqrt{40/2} = 4.47$ times in linear scale). The GO-coded optical pulse sequence measured at the fibre input with a duty cycle of 20% is represented by the blue curve in Fig. 3a. Note that the actual peak power of the first pulse has been adjusted equal to that of optimised single pulse for a fair comparison, and is here normalised to 1 in the figure for the sake of clarity. The corresponding $d_f(n)$ used for decoding is precisely retrieved based on the actual envelop of this sequence, as the red curve shown in Fig. 3a. Using such $d_f(n)$ and based on Eq. (3), decoded single-pulse responses for each scanning frequency are obtained, with a total decoding time of 1.8 s that has no critical impact on the total measurement time (~2.8 min).

The decoded single-pulse response at the Brillouin resonance is shown by the red curve in Fig. 3b, together with BOTDA traces obtained from the single-pulse scheme using 1024 (blue) and 17795 (black) averages, respectively. The latter case is set as a reference, whose SNR improvement with respect to the 1024-averaged single-pulse measurement is equivalent to the one that

could be ideally obtained by the designed GO-code scheme ($10lg\sqrt{17,795/1024} = 6.2$ dB). The figure demonstrates that the decoded response remains undistorted compared to the reference trace, validating the adaptability of the GO-code scheme to an uneven pulse power distribution along the sequence. Due to the additional Brillouin-gain-dependent optical noises in the GO-coded system (see Supplementary Note 3), a clear difference in the noise behaviour between the decoded response and the reference trace (visualised through the thickness of all traces) can be observed, as quantified by the SNR profiles shown in Fig. 3c. The impact of these noises reduces with distance due to the fibre attenuation, and eventually becomes negligible (i.e. showing a tiny impact on the expected coding gain) near the fibre far-end thanks to the proper optimisation of $F_E$. Although this behaviour has not yet been clearly documented, it is evident that such enhanced Brillouin-gain-dependent optical noises are inherent to any coded-BOTDA, being a non-exclusive effect in the GO-code proposed here, as experimentally verified for Simplex-coded BOTDA in Supplementary Fig. 9.

From the measured Brillouin loss spectrum, the Brillouin frequency shift (BFS) profile along the sensing fibre is estimated through a cross-correlation method[67]. The obtained BFS profile around a 5-m long hotspot placed at the fibre far-end is shown by the black curve in Fig. 3d, which matches well with a reference curve (blue) obtained from the single-pulse scheme with 2 m SR and 17,795 averages, and outperforms the single-pulse measurements with 1024 averages (red). To eventually verify the performance of the proposed GO-code, the BFS uncertainty profiles are calculated over five consecutive measurement for both single-pulse (blue) and GO-code (red) schemes, as shown in Fig. 3e. The frequency uncertainty at the fibre far-end obtained by the GO-code scheme (0.63 MHz) compared to the single-pulse scheme (2.65 MHz) is 4.1-fold enhanced, in good agreement with the theoretically evaluated coding gain (6.2 dB).

### Table 2 Parameters setting.

**Parameters setting of DGA**

Number of subpopulations $\alpha = 60$
Number of individuals in each subpopulation $\beta = 60$
Crossover probability $P_c$: random value between 0.8 and 0.9
Bit number of alternating segments $N_u^c = 19$
Mutation probability $P_m$: random value between 0.2 and 0.4
Bit number of mutation $N_u^m = 21$
Migration interval $\varphi = 60$
Number of migrants $\gamma = 12$
Generation counter $\mu = 60$

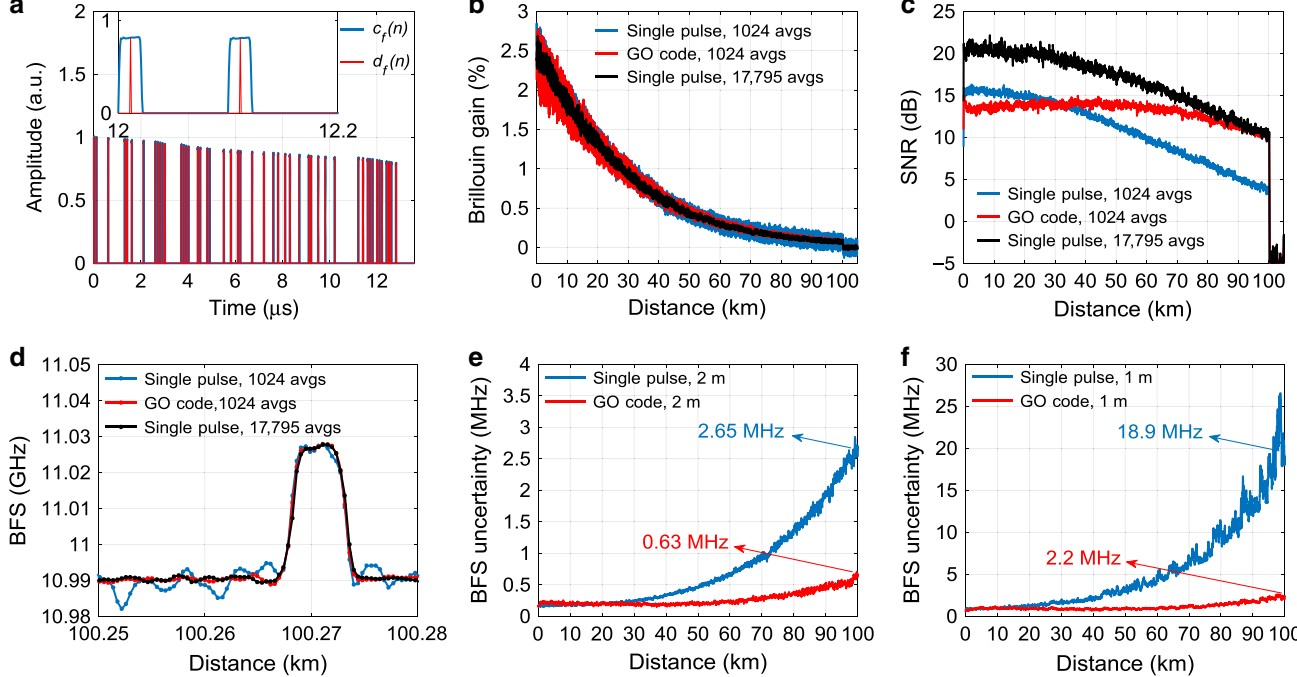

**Fig. 3 Experimental results of GO-coded and single-pulse BOTDA. a** A 136-bit normalised coding sequence $c_f(n)$ measured at the fibre input (blue) and the retrieved $d_f(n)$ (red). Inset: Zoom-in of normalised $c_f(n)$ and $d_f(n)$ over a time span from 12 to 12.2 μs. **b** Temporal BOTDA gain traces at fibre Brillouin resonance. **c** SNR profiles over the entire sensing fibre. **d** Measured BFS profiles around a 5-m-long hotspot located near the fibre far-end. **e, f** BFS uncertainty profiles along with the sensing fibre for 2 and 1 m SRs, respectively.

A similar comparison is then carried out at 1 m SR, which allows for a larger optimal energy enhancement factor $F_E = 200$ due to the smaller optimised single-pulse Brillouin gain (0.5%) at the fibre near-end. In this case a 723-bit GO-code sequence is found by DGA, offering a theoretically evaluated coding gain of 9.3 dB, being 0.7 dB lower than the reference coding gain. Whilst behaviours similar to the case of 2 m SR are observed (see Supplementary Fig. 13a–c), an 8.59-fold BFS uncertainty reduction (from 18.9 MHz for single-pulse scheme to 2.2 MHz for GO-code scheme) is demonstrated, as shown in Fig. 3f, matching perfectly the theoretically calculated coding gain of 9.3 dB.

**GO-coded ROTDR**. In this section, the performance of the proposed GO-code for ROTDR is investigated and compared with an optimised single-pulse case, both based on the same experimental layout (see "Methods").

Comparative measurements with 2 m SRs are carried out over a 39 km sensing range. A standard single-mode fibre (SMF) is used to eliminate any multimodal dispersion, thus avoiding SR impairments near the fibre far-end[1]. For both single-pulse and GO-code schemes, each temporal trace is acquired over 0.408 ms and averaged 2 million times, leading to a practical measurement time of 13.6 min. In the GO-code scheme, RZ format with a duty cycle of 22.2% is used to avoid the interaction between pulses in the code sequence and their forward Raman scattering components[57]. The energy enhancement factor $F_E$ for the GO-code ROTDR scheme is limited by the contamination of the amplified shot-noise (as elaborated in the Supplementary Note 4), and its optimal value is calculated to be $F_E = 44$. Taking into account the decaying envelop of the pulse sequence imposed by the EDFA gain saturation, the DGA delivers a 177-bit GO-code with a theoretical coding gain $G_c = 6.37$ dB, being only 0.34 dB lower than the standard reference coding gain $G_r = 6.71$ dB. The normalised GO-coded pulse sequence measured at the EDFA output is shown by the blue curve in Fig. 4a, along with the reconstructed $d_f(n)$ (red curve) that is required for decoding. The decoding time is found to be 10 ms, showing a negligible impact on the total measurement time as well.

The resolved temperature profiles obtained from the single-pulse and GO-code schemes are shown by the blue and red curves in Fig. 4b, respectively, indicating the clear performance enhancement provided by the proposed GO-code. Such a performance enhancement is then quantified by computing the corresponding SNR profiles, as shown in Fig. 4c. A coding gain of 6.33 dB at the fibre far-end is demonstrated, being in good agreement with theory. As anticipated, the SNR improvement at the fibre near-end is lower than the theoretical value due to the signal-power-dependent amplified shot-noise in the avalanche photodetector (APD), as elaborated in Supplementary Note 4. To verify the SR achieved by the GO-code scheme with ROTDR, the temperature of a 5-m-long hotspot introduced at the fibre far-end has been increased up to 70 °C, while the rest of the fibre is kept at the room temperature (25 °C). The temperature profiles obtained using the GO-code scheme is shown by the red curve in Fig. 4d, matching well with the trend of a reference temperature profile (blue curve) obtained by the single-pulse scheme with 2 m SR. Finally, Fig. 4e demonstrates a 4.27-fold temperature uncertainty enhancement at the fibre far-end (from 8.3 °C in the single-pulse case down to 1.9 °C with GO-code) thanks to the GO-code coding gain.

Measurements with 1 m SR are then performed with the same sensing fibre and measurement time. In this case, the optimised energy enhancement factor $F_E$ is determined to be 150, leading to a 450-bit code that takes into account the decaying trend of the sequence envelops. The GO-code delivered by the DGA exhibits a coding gain of $G_c = 8.67$ dB. Relevant results associated with 1 m

SR are illustrated in Supplementary Fig. 14a–c. Benefiting from the coding gain, the GO-code scheme offers 3.9 °C temperature uncertainty at the fibre end, representing a 7.24-fold improvement with respect to the uncertainty of 28.2 °C obtained by the single-pulse scheme, as shown in Fig. 4f.

In addition to the advantage of simple implementation, the capability of fast temperature sensing enabled by the proposed GO-code is demonstrated with 2 m SR and a 10.2-km SMF, using the same 177-bit code as illustrated in Fig. 4a. Temporal traces are acquired over 0.12 ms and averaged 9000 times, leading to a practical measurement time of ~1 s and decoding time of 1.6 ms. It must be mentioned that this short measurement and decoding time is indeed an exclusive feature of the proposed code, and cannot be reached by any other conventional codes. Due to the APD-induced amplified shot noise, the SNR improvement at the fibre far-end is found to be 5.5 dB, as quantitatively shown by the SNR profiles at 10.2 km in Fig. 4c. This corresponds to a 3.5-fold temperature uncertainty reduction, as can be observed in Fig. 4e (at 10.2 km, the GO-code reduces the temperature uncertainty from 8.3 °C in the single-pulse case down to 2.4 °C).

Based on these characterisations, a fast on-line real-time measurement that monitors the temperature of water under heating (from 25 to 100 °C) is performed. Figure 5a shows the two-dimensional map of the retrieved temperature as a function of the fibre position and the acquisition time, in which the temperature evolution of the hotspot (at 10.176 km) under heating can be identified. Figure 5b shows the measured evolution of the hotspot temperature (red) as a function of time, in good agreement with that measured by the single-pulse scheme (blue) and reference curve (black, measured by a high-precision thermometer). Figure 5c shows the temperature difference between measurements and reference, verifying the 3.5-fold performance enhancement. To explicitly show the fast decoding capabilities of the proposed GO-code scheme, a video is included in the Supplementary Material, displaying a real-time measurement of fast heating and cooling processes over two fibre sections (2 and 5 m, separated by a 5 m unperturbed section) at the fibre far-end. In the video, figures are obtained by on-line real-time decoding (synchronised with the measurements).

## Discussion

Whilst providing an SNR improvement comparable to sophisticated state-of-the-art techniques, the intrinsic mechanism of the proposed GO-code leads to overall advantages listed in Table 3:

1. the launch of only one sequence into the sensing fibre, minimising operation time and data storage. Note that Cyclic codes share the same advantage, while Simplex and Golay codes demand additional codeword switching time and larger memory usage, as indicated in the two top rows of Table 3;

2. a negligible post-processing (decoding) time (e.g. 1.6 ms decoding time with respect to the measurement time of 1 s for ROTDR), enabling real-time on-line fast sensing. The decoding time for each coding technique is summarised in the third row of Table 3, in which it can be found that the Cyclic coding has the fastest decoding process. For instance, in the case of a 100 km-long sensing range using a sampling rate of 250 MS/s and a 255-bit code (i.e. $N_h = 250{,}000$, $N_u = 255$), the evaluated decoding time of Cyclic is three times shorter than that of the proposed GO-code;

3. an aperiodic structure avoiding detrimental effects existing in single-sequence periodic (cyclic) codes, as indicated in the fourth and fifth rows of Table 3:

a. the impaired SNR all along with the sensing fibre due to the additional signal-dependent noise originating from the first

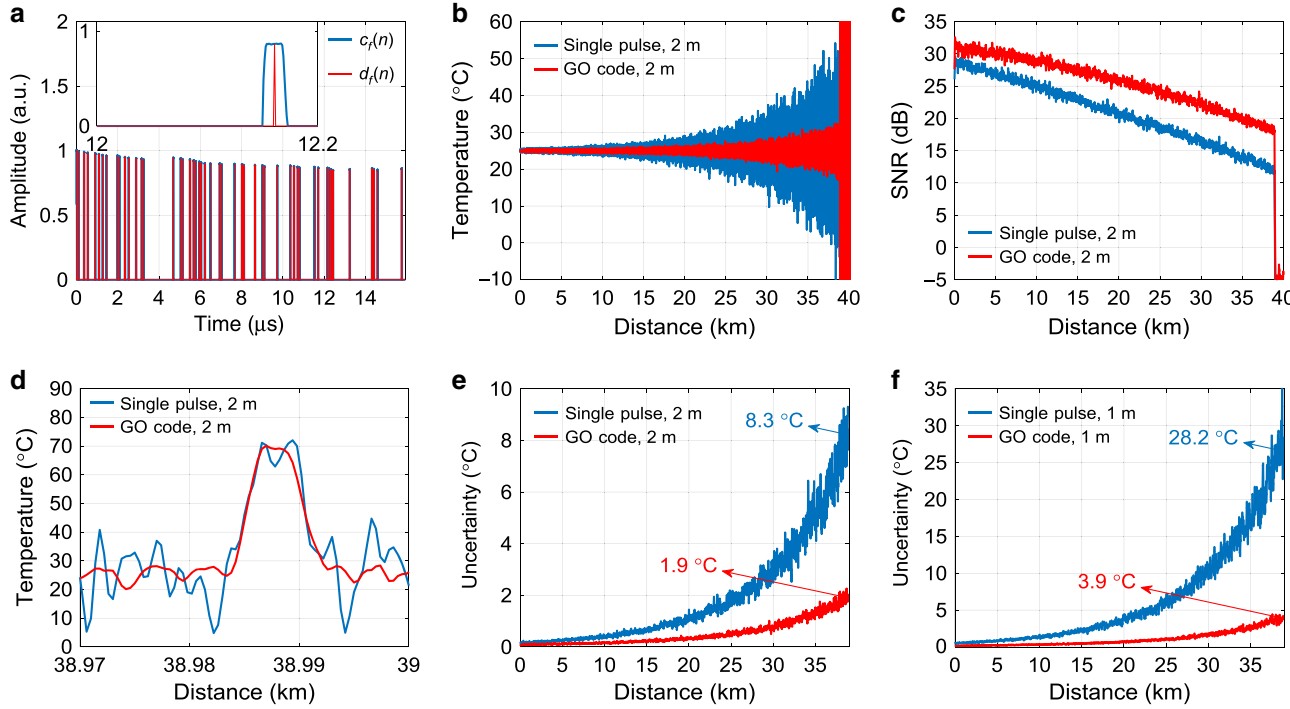

**Fig. 4 Experimental results of GO-coded and single-pulse ROTDR. a** A 177-bit normalised coding sequence $c_f(n)$ measured at the fibre input (blue) and the retrieved $d_f(n)$ (red). Inset: Zoom-in of normalised $c_f(n)$ and $d_f(n)$ over a time span from 12 to 12.2 μs. **b** Retrieved temperature profiles over the entire sensing fibre, for single-pulse (blue) and GO-coded (red) schemes. **c** SNR profiles over the entire sensing fibre, for single-pulse (blue) and GO-coded (red) schemes. **d** Retrieved temperature profiles around a 5-m-long hotspot located near the fibre far-end. **e**, **f** Temperature uncertainty profiles along with the sensing fibre for 2 and 1 m SRs.

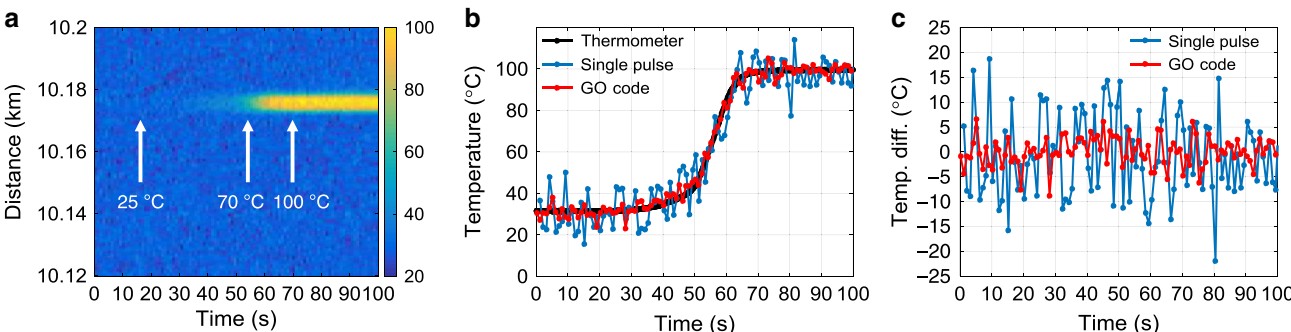

**Fig. 5 Experimental results of on-line real-time measurement for the temperature of water under heating. a** 2D map of the retrieved temperature as a function of time and fibre position. For the sake of clarity, the figure only shows fibre positions from 10.12–10.2 km. **b** Evolution of the retrieved temperature at the hotspot. **c** Error on the retrieved temperature at the hotspot as a function of time, obtained by subtracting the temperature measured by a thermometer to that from a single-pulse scheme (blue) and GO-coded scheme (red), respectively.

**Table 3 Performance comparison of different code types.**

| Code type | Cyclic | Simplex | Golay | GO-code |
|---|---|---|---|---|
| [a]Codeword switching time | 0 | $(N_u - 1)t_{cs}$ | $3t_{cs}$ | 0 |
| [b]Data storage (number of points) | $N_h$ | $N_u N_h$ | $4N_h$ | $N_h$ |
| [b]Decoding complexity | $O(2N_h\log_2 N_u)$ | $O(N_u^2 N_h)$ | $O(4N_h\log_2 N_h)$ | $O(2N_h\log_2 N_h)$ |
| Robustness to baseline fluctuations | × | √ | √ | √ |
| Tolerence to signal-dependent noises | × | √ | √ | √ |
| Tolerance to non-uniform code envelop | × | × | × | √ |
| Arbitrary $F_E$ required by a given system | × | × | × | √ |

[a]$t_{cs}$ is the time taken by the hardware to switch a code sequence to another when using a code type containing multiple coding sequences.
[b]Here, $N_r = N_h + N_c - 1 \approx N_h$ (since $N_h \gg N_c$ for a long sensing range) is assumed for Simplex, Golay and GO-code for the sake of simplicity and a more intuitive comparison.

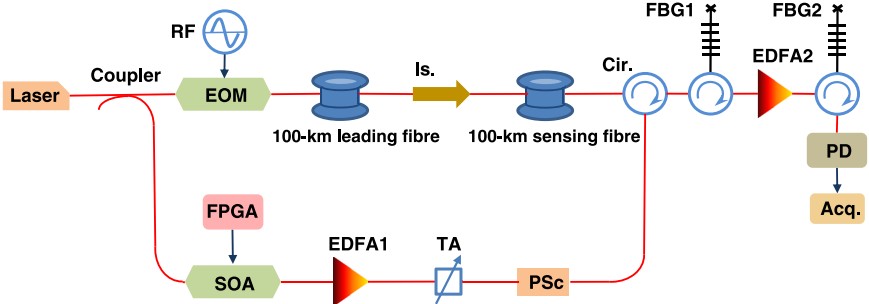

**Fig. 6 Experimental setup for both GO-code and single-pulse BOTDA.** RF radio frequency, EOM electro-optic modulator, Is. isolator, Cir. circulator, TA tuneable attenuator, PSc polarisation scrambler, SOA semiconductor optical amplifier, FBG fibre Bragg grating, PD photodetector, Acq. acquisition.

kilometres of the fibre (theoretically justified in Supplementary Notes 3 and 4). This leads to a largely compromised coding gain at all fibre positions, as experimentally demonstrated using BOTDA in Supplementary Note 3;

b. the absence of the continuous background observed in the measured cyclic-coded response, hindering any signal normalisation[53] required before decoding (the signal normalisation is an essential process before decoding to extract the linear coded response from the measured data[53]);

4. an adaptability to a non-uniform power over the pulse sequence (e.g. resulting from EDFA gain saturation), as indicated in the sixth row of Table 3, thus avoiding any modification of the instrumental layout compared to an optimised single-pulse implementation. This represents the ultimate asset of the technique, in full contrast with Simplex and Golay codes that require additional devices (higher cost) or pre-distortion operation (more complexity) to alleviate the unevenness of the coded sequence imposed by the EDFA gain saturation. Even with an improved uniformity (the amplitude envelope of the sequence is never perfectly uniform), decoding distortions remain still present when using Golay codes[53], resulting in a degraded performance;

5. enabling the use of any number of pulses in the code sequence, thus securing the highest coding efficiency in any given condition.

Benefiting from all these advantages, the proposed GO-coding technique enables, for the first time to the best of our knowledge, a pure overall performance improvement over optimised conventional DOFS without any hardware modification. The method only requires a software modification for sequence generation and post-processing, providing a fully compatible and cost-effective solution to improve the performance of existing instruments.

The performance of the proposed GO-code has been experimentally demonstrated by comparing with fully optimised single-pulse BOTDA and ROTDR (i.e. using a maximised signal power[25,26,28], and minimised noise bandwidth by applying digital Gaussian filtering[31]). In the case of Raman sensing, further improvement in SNR could be achieved by replacing the sensing fibre by a few-mode fibre[1,68,69], which enhances the allowable peak power of the pump light, with a moderate penalty in the SR at medium-long distances.

It is worth pointing out that the SNR enhancement provided by the GO-code and its features of single-sequence and fast decoding can lead to a huge reduction of the measurement time while maintaining other specifications. For instance, the 8.7-fold SNR enhancement demonstrated in the implemented GO-coded BOTDA sensor with 1 m SR can be used to reduce the number

of traces averaging, leading to a measurement time reduction of factor 75 ($=8.7^2$), while keeping the same spatial and measurand resolutions as the single-pulse BOTDA implementation for the same sensing distance. This means that a single-pulse measurement of 5.7 min can be performed by a GO-coded system in only ~5 s, thus leading to a significant enhancement in the response time of the sensor with no penalty on other specifications.

It must be mentioned that the code sequences found by the DGA may not be optimal, which is a common feature shared in most searching algorithms[66,70]. In addition, due to the high computational complexity when searching long code sequences, the finite computational capability limits the searching efficiency. This means that the theoretical maximum coding gain indicated in Fig. 2a, which may even exceed the gain provided by conventional codes (so far only being observed in the case of small $F_E$ as shown in Fig. 2c), is still possible to be reached if the searching is carried out with more powerful tools and approaches.

Note also that, although the code studied here has been designed in unipolar format (involving 0's and 1's elements), the DGA can be readily adapted to search for codes in bipolar format (involving −1's and 1's elements) that can be applied to phase-encodable DOFS, such as phase-sensitive optical time-domain reflectometers (φ-OTDR)[1,2]. Furthermore, the presented approach can benefit not only DOFS but also diverse fields in technical science where the retrieval of the single-pulse response with high SNR is required, such as incoherent OTDRs for fault detection, laser range finders, lidar systems, among others.

## Methods

**BOTDA setup.** The experimental configuration used for GO-coded BOTDA is shown in Fig. 6, where the hardware layout is strictly the same as that for a standard single-pulse BOTDA. In detail, a continuous-wave (CW) light from a distributed feedback laser operating at 1551 nm is split into probe and pump branches by a 50/50 coupler. In the probe branch (upper arm in the figure), the light travels through an electro-optic modulator driven by a microwave signal, producing a carrier-suppressed double sideband modulated probe wave. The frequency of the driving radiofrequency signal is scanned to reconstruct the BGS along the sensing fibre. This dual-sideband probe is launched into a 200 km-loop configuration[63] with an optical power of 3 dBm/sideband. This loop configuration consists of two fibres: the 100 km-long leading fibre delivers a probe wave with −18 dBm/sideband to the far-end of the other 100 km-long sensing fibre that acts as a distributed sensing element, thus realising a real remoteness of 100 km for the sensing range[63]. An isolator is placed between the leading and sensing fibres to block the pump wave, thus restricting the Brillouin interaction only to the sensing fibre. In the pump branch (lower arm in the figure), the light is intensity-modulated to a single optical pulse or a coded optical pulse train, using a high extinction ratio (>50 dB) semiconductor optical amplifier (SOA) driven by a commercial FPGA. Preceded by an EDFA and a tuneable attenuator to optimise the pump wave power to ~23 dBm (determined by the onset of MI[28]), a polarisation scrambler is inserted to mitigate polarisation fading and polarisation pulling effects[53].

In the receiver part, an EDFA operating in linear-gain regime is used between two tuneable narrowband fibre Bragg gratings (FBGs). The FBG1 selects the high-frequency probe sideband for the detection (i.e. making the system to operate in Brillouin loss configuration), thus preventing the EDFA from being saturated by

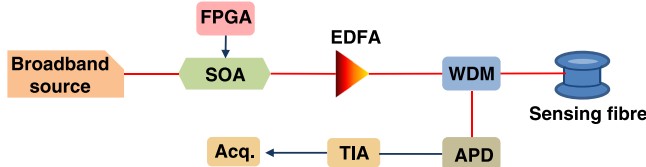

**Fig. 7 Experimental setup for GO-code and single-pulse ROTDR.** SOA semiconductor optical amplifier, WDM wavelength division multiplexer, APD avalanche photodiode, TIA transimpedance amplifier, Acq. acquisition.

unwanted frequency components, such as the Rayleigh backscattering and the second probe sideband. The FBG2, having the same filtering window as that of the FBG1, is used to filter out the amplified spontaneous emission (ASE) noise introduced by the EDFA, and thus to mitigate the ASE–ASE beating noise at the photo-detection. The selected optical signal is finally detected by a 125 MHz photodetector and acquired by a real-time data acquisition system with a sampling rate of 250 MS/s.

**ROTDR setup.** Similar to the BOTDA case, the experimental configuration used for GO-coded ROTDR is strictly the same as that for standard single-pulse ROTDR, as shown in Fig. 7. A CW light from a broad-linewidth (1.2 nm) light source operating at 1551 nm is injected into a high extinction ratio (>50 dB) SOA, driven by a commercial FPGA, to modulate the light intensity with a single optical pulse or a GO-coded pulse sequence, alternatively. The lightwave is then amplified by a high-gain EDFA to a peak power of 1 W and launched into the sensing fibre through a wavelength division multiplexer (WDM). The backscattered spontaneous anti-Stokes (AS) Raman signal is extracted by the same WDM and sent to an APD with a bandwidth of 125 MHz followed by a transimpedance amplifier. The photoelectrical signal is acquired by a real-time data acquisition card with a sampling rate of 250 MS/s.

Note that such a layout corresponds to a simplified ROTDR setup that makes use of a single APD and acquisition channel to measure only the AS spontaneous backward Raman scattering. This simplified setup allows us carrying out a proof-of-concept experiment in lab-controlled conditions. Wavelength-dependent losses are made negligible, so that the Raman Stokes or Rayleigh component is not required for real-time compensation. Instead, a simple pre-calibration procedure that measures the precise AS intensity at room temperature is performed to compensate for local and distributed losses along the sensing fibre.

**Distributed genetic algorithm.** Genetic algorithms (GAs)[70] are efficient search and optimisation techniques based on the principle of natural genetics evolution, aiming at finding optimum or sub-optimum solutions to problems that are difficult to be algebraically solved. Standard GAs perform the search through a search space involving a population of individuals, each of which is randomly set following predefined initial conditions and represents a potential solution for the given problem. During the search process, the algorithm iteratively evaluates the fitness value of each individual and breeds new individuals by carrying out operations found in natural genetics. Once given termination conditions are satisfied, the searching process stops, and the individual exhibiting the best fitness value is output as the solution. The key idea of GAs is that the new generation of population bred after each iteration should involve individuals representing better solutions thanks to the use of classical operators inspired by the advancement in nature. These operators consist of: (1) *selection*, which selects the potential parents of offspring based on the evaluation of fitness value in such a way that better individuals are often selected for matching; (2) *crossover*, in which the genetic information from two parent individuals are exchanged to generate two offsprings; and (3) *mutation*, which creates new genetic information that does not exist in the parents. In order to expand the searching scope and to accelerate the search speed, DGAs[66] that process multiple populations in parallel are often used. Furthermore, by periodically exchanging individuals among subpopulations via a *migration* operator, the premature convergence problem existing in conventional GAs can be greatly alleviated.

In this work a dedicatedly designed DGA is adapted to search for the optimal/ sub-optimal unipolar binary code sequence $u(n)$ that can provide the smallest possible noise scaling factor $Q$ (the largest possible coding gain $G_c$) for these given input parameters:

1. the energy enhancement factor $F_E$ bound by the physical constraints, as elaborated in Supplementary Notes 3 and 4;
2. the value of $m$, which is a number between 3 and 4 according to the empirical results shown in Fig. 2b;
3. the estimated envelope function $f'(n) = u_f(n)/u(n)$ imposed by the EDFA saturation, which can be estimated from the selected $m$ and EDFA specifications. Note that $f'(n)$ is close to the $N_x$-point downsampling of $f(n)$, the latter being rigorously retrieved from the measured $c_f(n)$.

The pseudo-code for the designed DGA is sketched in Algorithm 1 and elaborated in the following, which basically consists of three sections: initialisation (line 1–2), genetic processing (line 4–10) and stopping criteria (line 3, 11–17).

Firstly, $\alpha$ subpopulations denoted as $S_i (i = 1, 2, …, \alpha)$ are defined, each of which contains $\beta$ $N_u$-bit-long all-zero sequences (individuals) that are designated as $u_{i,j} (i = 1, 2, …, \alpha; j = 1, 2, …, \beta)$, where $N_u = mF_E$. Then the initialisation is performed by randomly replacing $F_E$ '0' bits by '1' bits for each $u_{i,j}$, and the corresponding noise scaling factors $Q_{i,j}$ are calculated according to Eq. (7). Then all subpopulations are processed in parallel by iteratively applying $\varphi$ times the following genetic operators:

*Selection:* The selection operator updates a given subpopulation $S_i$ by selecting individuals based on the criterion that sequences with smaller $Q$ (as defined by our target) should have a higher chance to survive. To do so, a typical roulette wheel selection method[70] is implemented, where the survival probability (i.e. the probability passing to the next generation) of each sequence $u_{i,j}$ in a certain subpopulation $S_i$ is expressed as:

$$P\left(u_{i,j}\right) = \frac{\frac{1}{Q_{i,j}}}{\sum_{j=1}^{\beta} \frac{1}{Q_{i,j}}}, \qquad (7)$$

from which it can be found that the smaller the $Q_{i,j}$, the larger the $P(u_{i,j})$. Using such a selection algorithm, the subpopulation is updated by carrying out the following selection steps for $\beta$ rounds: (i) generate a random number $P_r$ between 0 and 1, (ii) add $P(u_{i,j})$ to the partial sum $\sum_j P(u_{i,j})$, starting from the first sequence $u_{i,1}$ of the given subpopulation till the condition $\sum_j P(u_{i,j}) \geq P_r$ is satisfied, and (iii) select the first sequence that makes $\sum_j P(u_{i,j})$ exceeding $P_r$ as the survival individual for the current round.

*Crossover:* The crossover operator is analogous to gene recombination, which produces offsprings by exchanging genetic material (i.e. alternating segments of code sequences) of the parents. The length of alternating segments is here set as $N_u^c$ bits and one-point crossover[70] is applied, in which two sequences in a certain subpopulation are randomly chosen and their $N_u^c$-bit-long segments starting from a random crossover point are swapped, generating two new offsprings. This process is repeated by $\beta P_{c_i}$ times for a certain subpopulation $S_i$, where $\beta$ is the number of individuals (sequences) and $P_{c_i}$ is the probability of performing the crossover action.

*Mutation:* Mutation is an occasional tweak in the bit value of a sequence in order to maintain and introduce diversity in the subpopulation. This is usually applied with a low probability $P_m$, because if the probability is set very high, the GA would regress to an undesired random search. For the unipolar binary code sequence in our proposal, mutation is carried out by flipping the value of $N_u^m$ randomly chosen bits of a sequence, which is repeated by $\beta P_{m_i}$ times for a certain subpopulation $S_i$.

After finishing these $\varphi$-round iterations, the migration operation is subsequently performed to exchange the information among updated subpopulations. This is realised by transferring $\gamma$ sequences with smaller $Q$ in a source subpopulation to replace the same number of sequences with higher $Q$ in a target subpopulation. Here each subpopulation $S_i$ is in turn defined as the source of the neighbouring subpopulation $S_{i+1}$, for $1 \leq i < \alpha$, while $S_\alpha$ is the source subpopulation of $S_1$. This way the information of 'good' sequences is spread among subpopulations; furthermore, the immigrants (i.e. $\gamma$ sequences) would effectively change the fitness landscape thus avoiding the potential problem of premature convergence.

In parallel with the migration operation, the sequence with minimum $Q$ ($Q_{min}$) among all subpopulations is selected as the best sequence $u_{best}$ for the current generation (line 11 in Algorithm 1). The searching process is terminated when the converging condition is satisfied, i.e. when there has been no improvement of $u_{best}$ for $\mu$ consecutive generations. This is indicated by a counter that is initialised to zero and counts the number of the generations for which there has been no improvement of $u_{best}$. As shown in lines 12–16 of the pseudo-code, the counter increases every time a better $u_{best}$ is not generated, which is however reset to zero if $u_{best}$ is updated and $\sum_{n=1}^{N_u} f'_d(n) u_{best}(n) \approx F_E$. The algorithm terminates when the counter value reaches $\mu$ and the $u_{best}$ of the last generation is output as the final solution.

## Algorithm 1

The distributed genetic algorithm.

**Input:** energy enhancement factor $F_E$, total bit number of a sequence $N_u = mF_E$, estimated envelop function $f'$, number of subpopulations $\alpha$, number of individuals in each subpopulation $\beta$, crossover probability $P_c$, bit number of alternating segments $N_u^c$, mutation probability $P_m$, bit number of mutation $N_u^m$, migration interval $\varphi$, number of migrants $\gamma$, generation counter $\mu$.

**Output:** the best sequence $u_{best}$ and the minimum noise scaling factor $Q_{min}$.

**1 Randomly initialise $\alpha \times \beta$ sequences $u$, where $\alpha$ is the number of subpopulation and each subpopulation contains $\beta$ sequences (individuals)**

2 $u_f = f'u$, and evaluate the noise scaling factor $Q$ for each $u_f$ by Eq. (7);

**3 while** *counter* $< \mu$ **do**

**4 for each subpopulation $S_i (i = 1, 2, …, \alpha)$ do** $\varphi$ **times**

5 $S_i \leftarrow$ ***Selection*** $(S_i, Q_{i,1}, Q_{i,2}, …, Q_{i,\beta})$;

6 $S_i \leftarrow$ ***Crossover*** $(S_i, P_{c_i}, N_u^c)$;

7 $S_i \leftarrow$ ***Mutation*** $(S_i, P_{m_i}, N_u^m)$;

**8 Update the noise scaling factor $Q_{i,1}, Q_{i,2}, …, Q_{i,\beta}$ based on $u_f = f'u$ and Eq. (7);**

**9 end for**
**10** $(S, Q) \leftarrow$ *Migration* $(S, Q, \gamma)$;
**11** $(u_{best}, Q_{min}) \leftarrow (S, Q)$;
**12 if** $u_{best}$ **is updated and** $\sum_{n=1}^{N_u} f'(n) u_{best}(n) \approx F_E$, **then**
**13** $counter \leftarrow 0$;
**14 else**
**15** $counter \leftarrow counter + 1$;
**16 end if**
**17 end while**

Note that the size of subpopulation, the parameters of genetic operations and the termination condition are all adjustable when using DGA. It has been verified experimentally that the proposed DGA with appropriate parameters is efficient in the search of optimal code sequence. Benefiting from the robustness of DGA, it is not necessary to tune all these parameters for a specific sequence length. All the numerical results in this paper are obtained with the single set of parameters shown in Table 2.

## Data availability

The source data files underlying Figs. 3–5 and Supplementary Figs. 6 and 8–14 that support the experiments of this study are available in Zenodo (https://doi.org/10.5281/zenodo.4028516).

## Code availability

The code of distributed genetic algorithm is deposited in Zenodo (https://doi.org/10.5281/zenodo.3909600).

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

## Acknowledgements

This study is supported by National Natural Science Foundation of China under Grant (61875018) and National Key R&D Programme of China under Grant (2017YFB0405500). M.A. Soto acknowledges the support from AC3E ANID-Basal Project FB0008. The authors thank Jianqi Hu from EPFL for the fruitful discussion on the coding optimisation.

## Author contributions

X.H. conceived the concept of deconvolution. Z.Y. and S.Z. developed the concept, established the theoretical model and designed the methodology. X.S. and Z.Y. developed the distributed genetic algorithm. X.S., Z.Y., X.H. and S.W. contributed to all experiments and analysis. M.A.S contributed to Raman experiments and background discussions. X.G. contributed to Brillouin noise analysis. X.S. and Z.Y. wrote the paper. M.A.S., X.H. and L.T. revised the paper. X.H., J.W. and L.T. supervised the project.

## Competing interests

The authors declare no competing interests.
