## [Peer Review File · Nature Communications]

REVIEWER COMMENTS

Reviewer #1 (Remarks to the Author):

Distributed optical fiber sensors (DOFS) represent a rapidly growing area of research and applications. The performance metrics of DOFS, such as spatial resolution, range, experimental uncertainty and acquisition duration, are all determined by the signal-to-noise ratio (SNR). The SNR, in turn, is limited by fundamental considerations: signal cannot be made any stronger due to the onset of competing nonlinear effects, and detection noise cannot be avoided. SNR is therefore often improved through tedious averaging over many repeating acquisitions. The number of repetitions may be alleviated, however, through coding of optical signals with carefully constructed sequences. Coding in DOFS are studies since the 1980's [42], and the went through a massive resurgence in the last decade (much due to the efforts of the authors and their collaborators). Despite large efforts, however, the use of codes in DOFS is still not optimized. In this work, the authors take large steps in that direction. The authors formulate a target criterion for the construction of "good" sequences, in terms of the noise spectrum of the sequence following filtering by the response of a specific DOFS system. The number of pulses in the sequence is optimized based on noise considerations. Next, upper bounds for the attainable coding gain are established, based on the number of pulses and their spreading over time. A genetic algorithm is used to identify candidate sequences which come close to the performance upper bound. The synthesis procedure avoids inefficient exhaustive searches and converges within reasonable time and resources. Last but not least, the anticipated coding gain is demonstrated experimentally for two common DOFS setups: Brillouin-optical time-domain analysis (B-OTDA) and Raman-optical time-domain reflectometry (R-OTDR). Coding gain approaching nearly 10 dB is demonstrated experimentally.

Coding gain has been demonstrated in DOFS systems before. The main added values I find in this particular work are the following: 1) Synthesis of sequences that takes into account the specifics of DOFS systems, such as the response of erbium-doped fiber amplifiers or in optimizing the number of pulses with respect to noise sources; 2) Formulation of performance upper bounds; and 3) Realistic design protocols for approaching those upper bounds. I expect that the work will raise great interest within the optical fiber sensors community, for which it is directly applicable. Moreover, I expect that the work would also be appealing to broader research audience in diverse subjects such as signal processing, sequence design and genetic algorithms. I therefore recommend that a suitably revised manuscript is published eventually in Nature Communications.

Prior to publication, however, I recommend that the authors address the following issues to make their manuscript better:

- The mathematical derivations in the main text and the supplementary information are not easy to follow. I did not find anything wrong with them, but they do not provide much intuition. Many symbols are in use, and it's hard to keep track of them all. In particular, it was difficult for me to figure out what is meant by the Energy Enhancement Factor F_E , and the ratio m which is a key design parameter. The authors could simply state the sequence has F_E pulses with a value of '1', and it is padded with zeros to reach a length of $m \cdot F_E$. The longer length is required to achieve freedom of design for control of the filtered noise spectrum. (I hope that my above understanding in correct... This is not easy).
- Immediately following Fig. 1 in the main text: is the terminology $f(n) \cdot d(n)$ correct? Shouldn't it stand for convolution rather than multiplication? I'm a bit lost here.
- I could not follow Supplementary Figure 3 and the arguments leading to Supplementary Eq. (13).
- In Supplementary Eq. (18), please state that the value F_E is rather large.
- In the caption of Supplementary Fig. 6: Is the pump power really -35 dBm? Isn't it +35 dBm?
- In Supplementary Fig. 7(b), I could not identify why and how F_E of 40 pulses represents an optimum
- In Supplementary Fig. 10(b), I could not conclude that noise at the far end of the fiber is dominated by the thermal noise contribution.

Reviewer #2 (Remarks to the Author):

This paper report on a novel technique of coding scheme optimised through a distributed genetic algorithm

(DGA) which can be used to enhance the performance in distributed fiber sensors.

Optical coding schemes are a well known technique used for enhancing fiber sensor performance, and this paper proposes an optimization of aperiodic coding scheme overcoming some limitations affecting other existing codes.

The paper is well written and the degree of novelty seems adequate for Nature Communications.

Some modifications would be required for the paper to be suitable for publications:

- The genetic optimization algorithm is not adequately described in the paper but only in the supplemental material. It is strongly advised to add a description of the DGA also in the main paper text.
- In the paper the performance comparison (graphs, tables etc) has been carried out for ROTR and BOTDA comparing GO coding with single pulse sensing schemes only. Comparison tables and/or graphs should also be provided comparing GO algorithm with respect to other sensing coding schemes such as Simplex/Golay etc. for same coding gain values, taking into account also computing time, required storage etc. in order to provide a fair comparison /synoptic table clearly showing to the reader the benefits of proposed algorithm with respect to existing schemes.

Reviewer #3 (Remarks to the Author):

The major claims of the paper are related to the advantages of the proposed genetic-optimised code in terms of:

- 1) Operation time and data storage
- 2) Hardware complexity
- 3) Coding efficiency
- 4) Post-processing requirement and real time fast sensing
- 5) Robustness to signal dependent noise sources

We agree that the proposed genetic-optimised coding technique is very interesting and effective in making the use of coding applicable to distributed sensing in real-world applications. However the concept of coding and their applications to real problems is well known, also based on a cyclic single sequence coding, and also several industrial products exploit their potential in ROTDR and not only.

For this reason this paper, although technically sound, convincing and well written lacks of sufficient novelty.

More specifically we have the following comments:

The authors proposed a variant of the general method of enhancing SNR of measurements by using optical pulse coding in BOTDA and RDTs schemes. The added novelty is that the approach is based on the use of a convoluted sequence to probe the fiber and deconvolution operation to retrieve the single pulse equivalent, while the codes are generated using a distributed genetic algorithm. Although not inherent to the proposed method itself, the authors also perform fast decoding using FFT.

While offering the additional benefits of ability to use non-uniform pulse intensity and arbitrary word length, their method is similar in essence to previously demonstrated coding schemes. In addition, the technique also faces issues of tradeoff between energy enhancement factor and coding gain and the complexity of the genetic algorithm limits scalability to long code length values offering larger coding gain.

1) In the abstract the authors write:

“ Here, a novel technique is proposed, encoding the interrogating light signal by a single-sequence aperiodic code and spatially resolving the fibre information through a fast post-processing. The code sequence is once [for ever] computed by a specifically developed genetic algorithm, enabling for the first time a remarkable performance enhancement using an unmodified conventional configuration for the sensor.”

But the claim that the decoding sequence is computed once forever is not new and the same is true for the claim on the needs not to change the configuration. Other commonly used formats such as cyclic simplex also have the same characteristics. It seems the main novelty is in the flexibility of the codes in terms of

arbitrary length of codeword with no restricting rules and aperiodicity which addresses gain saturation of EDFAs, which should have been the main message of the abstract.

2) In page 2, the authors state

"Note that $DF(k)$ is a known function, since $df(n)$ can be precisely and easily obtained by retrieving the energy of each optical pulse in the optical code sequence $cf(n)$ measured at the fibre input (see middle row of Fig. 1)."

Does this mean that accurate measurement with their scheme requires characterizing this function for each measurement session? What about the uncontrolled variations of the intensity of the pulses, which determine $df(n)$ in a generic interrogating scenario? Does not the error introduced in the decoding process due to this issue affect the measurement resolution? Please elaborate this.

3) In the introduction the authors state

"...Simplex codes and their derivatives, have been mostly used for DOFS. Both codes can provide SNR enhancement with the same number of total acquisitions as in single-pulse DOFS; however, the additional operation time (e.g. the decoding time and the codeword switching time) and the larger occupied memory (for storing distinct coded traces) compromise the coding operation efficiency."

It is not clear what they mean by codeword switching time. Can't

the codeword matrix for any given codeword also be constructed once and reused for subsequent measurements without performing this operation each time? And since the codewords are known, perhaps this could not be an issue. Please either explain this better or remove this statement.

4) Referring to Figure 2 authors say

"The optimum value of m for a given energy enhancement factor E is empirically found to be between 3 and 4, as illustrated in Fig. 2(b). The reason comes from the fact that the high computational complexity in the case of long code lengths (large values of E and m) leads to a reduction of the genetic optimisation efficiency."

This statement means that, since $m = Nu/FE$, even the new method whose main purpose is to address issues in other types of codes comes with an inherent trade-off between the energy enhancement factor and coding gain. This is also evident in the reported coding gains which are below 10 dB. The experimental results of BFS resolution 6.2 dB and 9.3 dB for 2 m and 1m spatial resolution measurements also confirm this.

5) In the discussion section the authors mention the non-optimality of the DGA and the computational complexity for long codewords and state:

"This means that the theoretical maximum coding gain indicated in Fig. 2(a), which may even exceed the gain provided by conventional codes (so far only being observed in the case of small FE as shown in Fig. 2(c)), is still possible to be reached if the searching is carried out with more powerful tools and approaches, e.g. quantum computation."

This statement further highlights the limitation in the applicability of the newly proposed method, and the suggestion of using quantum computation is hypothetical as it refers a technology which is not readily available.

6) Minor comments

Please correct wording

in abstract: "for ever.", below figure 2: "the here proposed code", page below figure 3, "here proposed GO code"...

Reviewer #4 (Remarks to the Author):

Overall, my opinion is that this is an excellent piece of work, with a number of interesting and (I believe) novel aspects.

The novel aspects include the use of a deconvolution technique, rather than a digital correlation, to unscramble the signals from the coded pulse sequence input to the sensing fibre. Although deconvolution techniques have been used in the past with DOFS, I am not aware of their being used in conjunction with a code sequence. This translates, as correctly pointed out by the authors, into the benefit of using a single code sequence. The design of the code sequence is also novel in this context, as is the analysis, although the latter does lean on Ref [53] in particular.

The paper is largely complete in that (including the Supplementary Section) it covers, the concept of the proposed approach, a detailed analysis of the limitations due to thermal noise and a variety of signal and probe-intensity related noise and distortion effects, the design of the code sequence, a practical demonstration with two separate distributed sensor types.

I found that most of the questions that came to my mind as I was reading the paper, and that I thought of raising in this review, were addressed later on.

The paper thus contributes to advances in the field of distributed sensors in a number of separate ways with a several new concepts and related analysis as well as practical demonstrations.

I think that the separation of the material between the main article and the supplementary information detracts from the readability of the work as a whole; however, I suspect that this separation is dictated by the journal's policies on length and structure.

I thought that the noise analysis in the supplement was particularly interesting (especially the BOTDA section) and as far as I am aware reveals novel aspects of the properties of noise build-up in pulse-coded BOTDA systems.

Before finalising the manuscript, the authors may want to consider and address the following points.

- a) The authors make the statement that time-domain methods are intrinsically best suited for long-distance sensing. I agree with this statement, but I think that the authors should articulate why they believe this to be the case, or reference a definitive source on the subject.
- b) The authors mention the issue of the distortion of the probe pulse sequence caused by insufficient energy storage (as population inversion) in the booster amplifier (EDFA 1 in Fig. 5). The deconvolution approach solves this problem but that is at the expense of a reduction of the peak power for pulses transmitted later in the sequence and, therefore a reduction in the total energy that is launched under optimised conditions. The authors do not discuss the option of pre-emphasising the code sequence to flatten the pulse-height distribution, or indeed of designing an amplifier with sufficient pump power to avoid, or at least alleviate this effect. In Ref [53] (by a subset of the authors of the article under review), the problem was alleviated by amplifying the light between the source and the EOM, so the reasons for moving away from this approach should ideally be explained. It may come down simply to a matter of experimental convenience, but in my view this point should be discussed.
- c) The authors state a receiver bandwidth of 125 MHz (and sampling rate of 250 MSPS); this bandwidth is well in excess of that required for even 1 m spatial resolution, let alone the 2 m resolution used in some experiments. Did the authors apply any filtering prior to acquisition or in the digital domain to their data?
- d) regarding Fig. 5, I could not see where the heated zone is located along the fibre; is it at the same location as Fig. 4 (d)? Actually, there are two Fig. 5 in the main paper – there appears to be a numbering error.

Once, again, I believe that this manuscript is a valuable contribution to the field and my recommendation is that it should be published, ideally with some comments from the authors on the points above.

Arthur H. Hartog

Thank you very much for your valuable suggestions about our manuscript, we sincerely appreciate the time you dedicated to critically evaluate our results and this will undoubtedly help us to improve the quality of our manuscript. We have thoroughly considered all your comments and revised our manuscript every time it is applicable. Our detailed responses to your specific questions are listed below.

Reviewer 1:

Distributed optical fiber sensors (DOFS) represent a rapidly growing area of research and applications. The performance metrics of DOFS, such as spatial resolution, range, experimental uncertainty and acquisition duration, are all determined by the signal-to-noise ratio (SNR). The SNR, in turn, is limited by fundamental considerations: signal cannot be made any stronger due to the onset of competing nonlinear effects, and detection noise cannot be avoided. SNR is therefore often improved through tedious averaging over many repeating acquisitions. The number of repetitions may be alleviated, however, through coding of optical signals with carefully constructed sequences. Coding in DOFS are studies since the 1980's [42], and the went through a massive resurgence in the last decade (much due to the efforts of the authors and their collaborators). Despite large efforts, however, the use of codes in DOFS is still not optimized.

In this work, the authors take large steps in that direction. The authors formulate a target criterion for the construction of “good” sequences, in terms of the noise spectrum of the sequence following filtering by the response of a specific DOFS system. The number of pulses in the sequence is optimized based on noise considerations. Next, upper bounds for the attainable coding gain are established, based on the number of pulses and their spreading over time. A genetic algorithm is used to identify candidate sequences which come close to the performance upper bound. The synthesis procedure avoids inefficient exhaustive searches and converges within reasonable time and resources. Last but not least, the anticipated coding gain is demonstrated experimentally for two common DOFS setups: Brillouin-optical time-domain analysis (B-OTDA) and Raman-optical time-domain reflectometry (R-OTDR). Coding gain approaching nearly 10 dB is demonstrated experimentally.

Coding gain has been demonstrated in DOFS systems before. The main added values I find in this particular work are the following: 1) Synthesis of sequences that takes into account the specifics of DOFS systems, such as the response of erbium-doped fiber amplifiers or in optimizing the number of pulses with respect to noise sources; 2) Formulation of performance upper bounds; and 3) Realistic design protocols for approaching those upper bounds. I expect that the work will raise great interest within the optical fiber sensors community, for which it is directly applicable. Moreover, I expect that the work would also be appealing to broader research audience in diverse subjects such as signal processing, sequence design and genetic algorithms. I therefore recommend that a suitably revised manuscript is published eventually in Nature Communications.

Prior to publication, however, I recommend that the authors address the following issues to make their manuscript better:

1. The mathematical derivations in the main text and the supplementary information are not easy to follow. I did not find anything wrong with them, but they do not provide much intuition. Many symbols are in use, and it's hard to keep track of them all. In particular, it was difficult for me to figure out what is meant by the Energy Enhancement Factor F_E , and the ratio m which is a key design parameter. The authors could simply state the sequence has F_E pulses with a value of '1', and it is padded with zeros to reach a length of $m \cdot F_E$. The longer length is required to achieve freedom of design for control of the filtered noise spectrum. (I hope that my above understanding in correct... This is not easy).

Response: We thank the Reviewer for the suggestion. The Reviewer's understanding regarding F_E and m is absolutely correct when the amplitude envelope of the coding sequence is perfectly uniform (the energy of each coded pulse is identical), so that the energy enhancement factor F_E is simply the number of pulses with value of '1'. However, the coding sequence practically presents a decaying envelope due to the EDFA gain saturation, in which the number

of pulses is not equal to (larger than) F_E . To cover all possible cases (uniform and non-uniform amplitude envelope), the only generic way is to define F_E as the summation of all elements in the coded sequence $u_f(n)$, expressed by $F_E = \sum_{n=1}^{N_u} u_f(n)$, which is less intuitive but rigorous. To make this point clearer, we have modified our presentation introducing F_E and m , as highlighted in red in the section of ‘Design and optimisation of the code’. In the modified version, we have clearly addressed ‘Note that F_E equals to the number of coded pulses only when the code operates with a uniform amplitude envelope (the case of other conventional codes)’ to provide more intuition and for a better understanding.

Regarding the lack of intuition in mathematical derivations as pointed out by the Reviewer, we have made following improvements in the revised version:

- 1) We have added two grey dashed curves in the middle row of Fig. 1 to visualize $f(n)$ (i.e. the amplitude envelope of the sequence), thus further emphasizing that we are dealing with a coding sequence with a decaying amplitude envelope.
- 2) We have added a sentence in the paragraph above Eq. (1), ‘Note that all sequences defined hereafter with subscript ‘ f ’ are associated with such a non-uniform amplitude envelop’, to provide more intuition and to keep better track of symbols.
- 3) Since most symbols are visualized in Fig. 1, we have added ‘see Fig. 1’ after describing relevant symbols in the main text to suggest readers referring to the figure for better intuition.

2. Immediately following Fig. 1 in the main text: is the terminology $f(n)*d(n)$ correct? Shouldn’t it stand for convolution rather than multiplication? I’m a bit lost here.

Response: We thank the Reviewer for highlighting this point. Here this should be definitely a multiplication rather than a convolution. We believe the misunderstanding by the Reviewer comes from the fact that we only mentioned $f(n)$ in the main text, but did not visualize it in any figure in the original manuscript. To make this point clearer, we have visualized $f(n)$ in the middle row of Fig. 1 by grey dashed curves. For more clarity, we have also modified the relevant sentence in the paragraph below Eq. 1 (please note that now the paragraph is no longer immediately following Fig. 1 due to the changed pagination).

3. I could not follow Supplementary Figure 3 and the arguments leading to Supplementary Eq. (13).

Response: We thank the Reviewer for the comment. As shown below, we found that in the original Supplementary Figure 3 the relation between the main figure and the inset was not separated clearly, which could lead to some confusion.

Original Supplementary Fig. 3

In the revised version, we have made a new Supplementary Fig. 3 as shown below for better intuition.

Revised Supplementary Fig. 3

We have also added descriptions to better explain Supplementary Eq. (13), which are highlighted in red in the revised version.

4. In Supplementary Eq. (18), please state that the value F_E is rather large.

Response: We thank the Reviewer for raising this important point. This is indeed a necessary condition for the Supplement Eq. (18) to hold. We have added relevant description right below Supplementary Eq. (18).

5. In the caption of Supplementary Fig. 6: Is the pump power really -35 dBm? Isn't it +35 dBm?

Response: The pump power written in the original manuscript is actually '~35 dBm' rather than '-35 dBm'. Regarding the confusion raised by the Reviewer, we have modified the relevant text from '~35 dBm' to 'around 35 dBm'.

6. In Supplementary Fig. 7(b), I could not identify why and how F_E of 40 pulses represents an optimum.

Response: We thank the Reviewer for the comment. As addressed in the manuscript, the upper bound value of F_E is limited by the signal-dependent noises that get more detrimental for larger F_E . Therefore F_E must be selected, so that it doesn't significantly enlarge the total noise with respect to that using a single-pulse scheme. We opt for the criterion that the optimum F_E must ensure less than 10% difference between the noise variances in coded and single-pulse cases. From Supplementary Fig. 7(b), we can observe that when F_E equals to 40, the above-mentioned noise difference amounts to 10%, so that F_E of 40 is considered optimum.

The confusion raised by the Reviewer may come from the vertical axis of the original Supplementary Fig. 7(b) that may not be straightforwardly grasped (the ratio between additional noise induced in coded scheme and the noise in single-pulse scheme). In the modified version, we have changed the vertical axis for a more direct approach (the ratio between the noises in coded and single-pulse cases), and modified the relevant descriptions in the paragraph below Supplementary Fig.7(b).

Original Supplementary Fig. 7(b)

Revised Supplementary Fig. 7(b)

7. In Supplementary Fig. 10(b), I could not conclude that noise at the far end of the fiber is dominated by the thermal noise contribution.

Response: We thank the Reviewer for the comment. We believe the confusion comes from the fact that we did not specify the length of the sensing fibre (39 km). In Supplementary Fig. 10(b) (in the revised manuscript, this figure becomes Supplementary Fig. 11(b) since we inserted other figures as required by other Reviewers), the fibre far-end actually corresponds to 35~39 km, and the sampling points after 39 km (i.e. 39~45 km) corresponds to the response outside the fibre. In a spontaneous scattering process like in the Raman sensor, there is no backscattered light from outside the fibre, so that it can be safely considered there is only a thermal noise contribution. Since the noise level at the fibre far-end is similar to this thermal noise, we can reasonably conclude that the signal close to the fibre far-end is dominated by the thermal noise contribution.

To make this point clear, we have added the fibre length information and modified the relevant description.

Reviewer 2:

This paper report on a novel technique of coding scheme optimised through a distributed genetic algorithm (DGA) which can be used to enhance the performance in distributed fiber sensors.

Optical coding schemes are a well known technique used for enhancing fiber sensor performance, and this paper proposes an optimization of aperiodic coding scheme overcoming some limitations affecting other existing codes.

The paper is well written and the degree of novelty seems adequate for Nature Communications.

Some modifications would be required for the paper to be suitable for publications:

1. The genetic optimization algorithm is not adequately described in the paper but only in the supplemental material. It is strongly advised to add a description of the DGA also in the main paper text.

Response: We thank the Reviewer for the suggestion and we fully agree with the Reviewer's opinion. In the revised manuscript, we have moved the detailed description of DGA and the associated pseudocode from Supplementary note 2 to Method section in the main text.

2. In the paper the performance comparison (graphs, tables etc) has been carried out for ROTR and BOTDA comparing GO coding with single pulse sensing schemes only. Comparison tables and/or graphs should also be provided comparing GO algorithm with respect to other sensing coding schemes such as Simplex/Golay etc. for same coding gain values, taking into account also computing time, required storage etc. in order to provide a fair comparison /synoptic table clearly showing to the reader the benefits of proposed algorithm with respect to existing schemes.

Response: We appreciate the Reviewer for the suggestion, making a fair comparison table can definitely highlight the advantages of the proposed code. In the revised manuscript, we have added a detailed comparison table in Discussion Section, and provided relevant descriptions right above the table. From the table, it can be easily concluded that the proposed code outperforms other existing codes in terms of all critical specifications.

Reviewer 3:

The major claims of the paper are related to the advantages of the proposed genetic-optimised code in terms of:

- 1) Operation time and data storage
- 2) Hardware complexity
- 3) Coding efficiency
- 4) Post-processing requirement and real time fast sensing
- 5) Robustness to signal dependent noise sources

We agree that the proposed genetic-optimised coding technique is very interesting and effective in making the use of coding applicable to distributed sensing in real-world applications. However, the concept of coding and their applications to real problems is well known, also based on a cyclic single sequence coding, and also several industrial products exploit their potential in ROTDR and not only.

For this reason, this paper, although technically sound, convincing and well written lacks of sufficient novelty.

Response: We respect the Reviewer's comments, and fully agree with the Reviewer that the concept of coding is well known and of course we are millions of miles away of claiming that the concept of coding is originally proposed by us. Instead, the major novelty of the proposed work is that it is the first approach fully realising the well-known coding concept, which has never been achieved with preceding methods, though intense research activities have been made by the community over past decades. In other words, the proposed technique maintains all advantages and overcome all detrimental issues of existing coding techniques, thus obtaining the highest coding efficiency that was only conceptual so far. We believe this represents sufficient innovation and a major progress. So we keep convinced with the best of our scientific and intellectual rigour that this approach is a real breakthrough in coding techniques and we feel sorry that our demonstration could not fully convince the Reviewer at the first reading. We hope that these comments and the revisions in the manuscript will raise all concerns from the Reviewer.

Even by making sincere efforts to be critically objective, we are still doubtful that the Cyclic code is a good candidate for distributed sensing, though it is also a single-sequence code featuring minimised data storage and code switching time. The major problem of the Cyclic code is that the actual coding gain (i.e. SNR improvement) is largely compromised (far lower than the theoretical coding gain) by signal-dependent noises, as elaborated hereafter.

It is important to grasp that the theoretical coding gain can be fully realised only if the noise level remains unchanged between coded and single-pulse schemes, which is an evidence and practically means that only the photo-detection thermal noise is relevant. However, it turns out that some signal-dependent noises (e.g. shot noise, signal-SpBS beating noise) that remains negligible in a single-pulse scheme, may however be significantly enhanced (even becoming dominant) in some coded schemes due to the largely enhanced pump energy. When the code is aperiodic, which is the case of the proposed code, Simplex code and Golay code, the energy of the entire coded sequence is much attenuated when reaching the fibre far-end, so that the detrimental impact of signal-dependent noises is largely alleviated (i.e., the actual coding gain at the fibre far-end results to be similar to the theoretical coding gain); however, when the code is periodic, which is the case of the Cyclic code, the coded pulses spread all along the fibre and the coded sequence always contains strong coded pulses at the fibre near-end, so that the noise induced by these strong pulses will impact on the SNR of the signal originating from weak pulses at the fibre far-end. This essentially means that signal-dependent noises remain large at all sampling points of the probe wave (i.e. all fibre positions). We are sorry to enter into such details, but this is of key importance for grasping the practical pros and cons of different types of coding.

We have addressed this question in the Introduction and Discussion (associated with citation [1]) in the original manuscript, but did not elaborate it since it is not the major scope of the paper. To draw the Reviewer's attention, here we specify that the point is briefly addressed on the page 95 of the cited book [1]: 'A. H. Hartog, Introduction to Distributed Optical Fiber Sensors, CRC Press, 2017), in the context of traditional OTDR, as shown by the screen shot below:

m-sequences have the property that their auto-correlation function consists of just $-1/m$ except at zero shift, or after a delay corresponding to a complete sequence duration, where it is 1. Thus, they provide an essentially perfect coding scheme. This approach was used to modulate the probe signal and then correlate the received backscatter signal with the same sequence [84]. Although a backscatter trace with correctly reconstituted spatial resolution was obtained, the improvement due to the coding was not clear.

More generally, the weakness of using continuous sequences such as m-sequences is that their output chip stream is also continuous. For spread-spectrum techniques, this means that the weak signals from the remote end arrive at the same time as strong signals from the near end. When the fibre is long, the signal will have a large dynamic range and so the signals returning from the remote end are orders of magnitude smaller than the near-end signal. In this situation, the strong near-end signals generate a large level of shot noise at the detector that is superimposed on the weak signal from the remote end. Therefore, beyond a certain dynamic range, the benefits of pulse-compression coding are negated because the remote signal is swamped by noise accompanying the strong near-end signals.

What is really required is a code having the good auto-correlation properties already discussed and a finite code length. No such codes exist for unipolar signals (such as optical pulse trains). Healey was probably the first to realise in this context [85] that a similar function could be achieved by using multiple codes that, *individually*, had unsatisfactory autocorrelation functions but when used *in combination*

In our original manuscript, we have thoroughly analysed the performance degradation due to signal-dependent noises in BOTDA and ROTDR, in Supplementary note 3 and 4, respectively. Although the established noise model is generic (can be used to evaluate any code type), demonstrations were focused on aperiodic code, this might be the reason why the Reviewer overlooked the detrimental issue in Cyclic codes. To solidly convince the Reviewer and to thank him for his collegial help, we have performed new experiments regarding the large SNR impairment when using Cyclic coding for BOTDA, and results are shown below.

Fig. r1. The standard deviations of theoretically predicted noise and measured noise as a function of distance for (a) single-pulse; (b) aperiodic coding and (c) Cyclic coding schemes.

For a fair comparison, the results shown in Figs. r1(a-c) are measured under the same experiment condition, in which the peak power of the first coded pulse is identical to the optimised single pulse power (just below modulation instability threshold). Between the 2 coding schemes, the number of coded pulses is both 40, corresponding to identical theoretical coding gain. It can be clearly seen that, with aperiodic code, the noise at the fibre far-end is similar to that of the single-pulse scheme, while the noise floor is globally higher than the single-pulse case when using Cyclic coding due to the spread signal-dependent noise as explained before. This means that the theoretical coding gain at the fibre far-end can be realised using aperiodic codes, as demonstrated by the SNR improvement shown in Fig. r2(a); however, in the case of Cyclic coding, there is nearly no actual coding gain, as shown in Fig. r2(b).

Fig. r2. Experimental SNR profiles obtained by single-pulse and coding schemes. (a) aperiodic code (b) Cyclic code. Both with optimised pulse peak power. (c) Cyclic code with 10 mW pulse peak power (far below the optimum value).

It must be mentioned that, the theoretical coding gain of Cyclic coding was realised and reported in the literature (e.g. [58]): this is because the pulse peak power in both single-pulse and Cyclic coding schemes were not optimised (10 mW, which is far below the modulation instability threshold). In this case, the signal-dependent noise is much lower, such that the actual coding gain is close to the theoretical value; however, it is of limited relevance since the absolute SNR with coding (red curve in Fig. r2(c)) turns out to be lower than that of an optimised single-pulse scheme as represented in the previous graph in Fig. r2(b) (blue curve). All results indicate that there is **no benefit** brought by Cyclic coding when comparing with a **fully optimised** single-pulse scheme. We keep ready to revisit this conclusion if an evidence can be brought that we skipped an essential point.

We have added the relevant figures and associated descriptions in the revised Supplementary note 3.

More specifically we have the following comments:

The authors proposed a variant of the general method of enhancing SNR of measurements by using optical pulse coding in BOTDA and RDTs schemes. The added novelty is that the approach is based on the use of a convoluted sequence to probe the fiber and deconvolution operation to retrieve the single pulse equivalent, while the codes are generated using a distributed genetic algorithm. Although not inherent to the proposed method itself, the authors also perform fast decoding using FFT.

While offering the additional benefits of ability to use non-uniform pulse intensity and arbitrary word length, their method is similar in essence to previously demonstrated coding schemes. In addition, the technique also faces issues of tradeoff between energy enhancement factor and coding gain and the complexity of the genetic algorithm limits scalability to long code length values offering larger coding gain.

Response: We fully agree with the Reviewer that the essence of the proposed code is similar to other existing code, since this is a coding and it follows the mathematical flow chart of coding. However, we must clarify as in the preamble of our response that the novelty of this work is to propose a new code fully realising all conceptual benefit of coding, while remaining of course a coding. Therefore we believe that there is a totally innovative approach in the conception of the coding process and it shows in essence a clear contrast with existing approaches.

Secondly, we cannot fully share the opinion of the Reviewer that ‘the technique also faces issues of tradeoff between energy enhancement factor and coding gain’. Indeed, there is no trade-off between energy enhancement factor and coding gain; instead, these 2 parameters are always positively linked, i.e., the larger the energy enhancement factor, the higher the coding gain. The ultimate limit of the energy enhancement factor (thus the coding gain) is the onset of undesired physical phenomena elaborated in [53] and the Supplementary notes 3 and 4. Note that this limitation is not specific to the proposed code (i.e. any technique cannot infinitely increase the pump energy for distributed sensing).

So, we agree with the Reviewer on the point that the proposed technique cannot exceed this physical limit as well, it is not magical. However, we would like to clarify that this is not opposed to the general and realistic methodology ‘under a given limitation, how to maximise the efficiency’. For instance, the SNR of a photodetection scheme has a well-known shot-noise limit, the scope of optimising the system is to reach this limit rather than overcoming it. Therefore we even consider to be ‘only limited by physical constraints’ is an advantage rather than an issue to overcome, even more if these limits are currently not reached by existing coding schemes.

Thirdly, like the Reviewer, it can be deemed at first glance that ‘the complexity of the genetic algorithm limits scalability to long code length values offering larger coding gain’. As addressed in the Supplementary notes 3 and 4, the physical constraints limit the energy enhancement factor to be no larger than 200. This means that we only require good coding sequences with energy enhancement factors below this value. As shown in Fig. 3(c) of the original manuscript, for an energy enhancement factor from 20 to 200, code sequences delivered by the genetic algorithm always show less than 0.4 dB SNR penalty, which cannot be even discerned in practical measurements. One may find higher energy enhancement factors in the literature addressing coding techniques, but this is simply because the single-pulse scheme used was not fully optimised (the pulse peak power was not maximised).

1. In the abstract the authors write:

“Here, a novel technique is proposed, encoding the interrogating light signal by a single-sequence aperiodic code and spatially resolving the fibre information through a fast post-processing. The code sequence is once [for ever] computed by a specifically developed genetic algorithm, enabling for the first time a remarkable performance enhancement using an unmodified conventional configuration for the sensor.”

But the claim that the decoding sequence is computed once forever is not new and the same is true for the claim on the needs not to change the configuration. Other commonly used formats such as cyclic simplex also have the same characteristics. It seems the main novelty is in the flexibility of the codes in terms of arbitrary length of codeword with no restricting rules and aperiodicity which addresses gain saturation of EDFAs, which should have been the main message of the abstract.

Response: We thank the Reviewer for the comment. We would like to clarify that it goes beyond our intentions to consider that we claimed or even suggested ‘the coding sequence is computed once forever is new and unique’. The only purpose of addressing ‘The code sequence is once forever computed...’ in the abstract is to clarify that whilst providing several other advantages, our code does not lose this common property shared by all existing codes.

Secondly, we must clarify that we never claimed ‘our code is the only one that does not need to change the single-pulse configuration.’ Our claiming point is that although some codes share the same property, e.g. Cyclic code, they face other serious issues. For instance, as addressed in the previous response, Cyclic code brings no SNR improvement when comparing to a fully optimised single-pulse scheme.

In short, we never intended to claim that all conventional codes have no good feature. Instead, our claiming point is that all conventional codes have their respective advantages and detrimental problems, while all advantages are kept and all issues are overcome using the proposed code, as summarized in the newly added Table 2 in Discussion. We believe that this is not a minor or incremental progress.

2. In page 2, the authors state

“Note that $DF(k)$ is a known function, since $df(n)$ can be precisely and easily obtained by retrieving the energy of each optical pulse in the optical code sequence $cf(n)$ measured at the fibre input (see middle row of Fig. 1).”

Does this mean that accurate measurement with their scheme requires characterizing this function for each measurement session? What about the uncontrolled variations of the intensity of the pulses, which determine $df(n)$ in a generic interrogating scenario? Does not the error introduced in the decoding process due to this issue affect the measurement resolution? Please elaborate this.

Response: We thank the Reviewer for the comment. We confirm that the decoding function needs to be characterised for a given experimental scenario. However, once all the experimental parameters (e.g. pulse width, code length, EDFA gain) are fixed, the characterization is not required for each measurement session. We don’t understand exactly what is ‘uncontrolled variation of the intensity of the pulses’ meant by the Reviewer, since according to our repeated experience the intensity of the pulses is well controlled in a well-designed system. If the Reviewer meant the intensity noises given by laser, SOA and EDFA, they do not distort the decoding function since the function is retrieved by measuring the pulse energy, where the intensity noises show a negligible impact.

Secondly, we don’t know what is the ‘error introduced in the decoding process’ meant by the Reviewer, since we never addressed any error in decoding. Actually one of the advantages of the proposed code is to be free of decoding errors, so there shouldn’t be any error. If the Reviewer meant the degraded coding gain, this is purely due to signal-

dependent noises as elaborated in Supplementary note 3 and 4, in which all theoretical models match perfectly with experimental results. In addition, such coding gain degradation also exists in other codes, so it can be safely considered that it is irrelevant to the decoding function characterisation.

3. In the introduction the authors state

“...Simplex codes and their derivatives, have been mostly used for DOFS. Both codes can provide SNR enhancement with the same number of total acquisitions as in single-pulse DOFS; however, the additional operation time (e.g. the decoding time and the codeword switching time) and the larger occupied memory (for storing distinct coded traces) compromise the coding operation efficiency.”

It is not clear what they mean by codeword switching time. Can't the codeword matrix for any given codeword also be constructed once and reused for subsequent measurements without performing this operation each time? And since the codewords are known, perhaps this could not be an issue. Please either explain this better or remove this statement.

Response: We thank the Reviewer for the comment, and apologize for the unclear statement. The ‘codeword switching time’ is the time taken by the hardware to switch a code sequence to another when using a code type containing multiple coding sequences (e.g. Simplex and Golay codes). Such switching time has negligible impact on the total measurement time if a large number of trace averages is performed, but can be an ultimate limiting factor when performing fast measurement (small average number, so the additional code switching time taken by the hardware dominates).

We have modified the relevant description to better explain the impact of the codeword switching time in the paragraph referred by the Reviewer, and explained it again in the footprint of the newly added Table 2 in Discussion.

4. Referring to Figure 2 authors say

“The optimum value of m for a given energy enhancement factor E is empirically found to be between 3 and 4, as illustrated in Fig. 2(b). The reason comes from the fact that the high computational complexity in the case of long code lengths (large values of E and m) leads to a reduction of the genetic optimisation efficiency.”

This statement means that, since $m = N_u/FE$, even the new method whose main purpose is to address issues in other types of codes comes with an inherent trade-off between the energy enhancement factor and coding gain. This is also evident in the reported coding gains which are below 10 dB. The experimental results of BFS resolution 6.2 dB and 9.3 dB for 2 m and 1 m spatial resolution measurements also confirm this.

Response: We thank the Reviewer for the comment, and we totally share the opinion that ‘even the new method has the same inherent limit’. However, this is not because ‘the proposed method has an inherent trade-off between the energy enhancement factor and coding gain’, as has been explained in a previous response. The energy enhancement factor F_E and the coding gain are actually positively linked. The reason of only achieving a coding gain of 6.2 dB and 9.3 dB for 2 m and 1 m spatial resolutions purely comes from the contamination of signal-dependent noises as mentioned before, which ultimately limit the maximum value of F_E .

As mentioned before, this physical limitation is shared by all code types. The aim of optimising a coding system is to reach such an ultimate limit. We would like to draw the Reviewer’s attention that only the proposed code can practically reach this limit thanks to the feature of an arbitrary length. For instance, if physical constraints determine a maximum energy enhancement of 40, for our proposed code one can use 40 coded pulses (1’s elements); however, for Golay codes one can only use around 32 pulses (1’s elements) because the total code length can only be 64 (its mathematical rule determines the code length to be 2^n , where n is an integer). In addition, whilst uniquely reaching such a physical limit, the proposed method simultaneously provides many other advantages as summarised in newly added Table 2 in Discussion.

Note that larger unipolar coding gains (> 10 dB) can be found in some of the literature addressing coding techniques, this is simply because the pulse peak powers in both single-pulse and coding schemes were not optimised (far below the modulation instability threshold).

5. In the discussion section the authors mention the non-optimality of the DGA and the computational complexity for long codewords and state:

“This means that the theoretical maximum coding gain indicated in Fig. 2(a), which may even exceed the gain provided by conventional codes (so far only being observed in the case of small FE as shown in Fig. 2(c)), is still possible to be reached if the searching is carried out with more powerful tools and approaches, e.g. quantum computation.”

This statement further highlights the limitation in the applicability of the newly proposed method, and the suggestion of using quantum computation is hypothetical as it refers a technology which is not readily available.

Response: We do not share the opinion of the Reviewer that ‘This statement further highlights the limitation in the applicability of the newly proposed method’. In the referred sentences we only intended to delivery a message that even better performance may be reached using more powerful computational tools, which does not mean the performance based on current computational tools is not sufficient.

As mentioned in previous responses, we only need to design codes with energy enhancement factor up to 200 (limited by physical constraints). Below this value, the penalty of coding gain due to limited computational power is always below 0.4 dB (see Fig. 2(c)) that can be considered negligible. In other words, with current standard computational tools, the proposed method has already achieved decent performance; more powerful computational tool serves as an added value that may give rise to a larger coding gain (cannot be infinitely larger, the ultimate value of coding gain is provided by Eq. (5) and illustrated in Fig. 2(a)).

But we fully share the opinion of the Reviewer that it is probably not appropriate to suggest quantum computation because we don’t have solid knowledge on that. In the revised version we have removed ‘e.g. quantum computation’. Thank you for suggesting and it will definitely make our paper more credible.

6. Minor comments

Please correct wording

in abstract: “for ever.”, below figure 2: “the here proposed code”, page below figure 3, “here proposed GO code”...

Response: We thank the Reviewer for pointing out these issues that skipped our attention. We have changed ‘for ever’ to ‘forever’, ‘the here proposed code’ to ‘the code proposed here’ in the revised manuscript everywhere applicable.

Reviewer 4:

Overall, my opinion is that this is an excellent piece of work, with a number of interesting and (I believe) novel aspects.

The novel aspects include the use of a deconvolution technique, rather than a digital correlation, to unscramble the signals from the coded pulse sequence input to the sensing fibre. Although deconvolution techniques have been used in the past with DOFS, I am not aware of their being used in conjunction with a code sequence. This translates, as correctly pointed out by the authors, into the benefit of using a single code sequence. The design of the code sequence is also novel in this context, as is the analysis, although the latter does lean on Ref [53] in particular.

The paper is largely complete in that (including the Supplementary Section) it covers, the concept of the proposed approach, a detailed analysis of the limitations due to thermal noise and a variety of signal and probe-intensity related noise and distortion effects, the design of the code sequence, a practical demonstration with two separate distributed sensor types.

I found that most of the questions that came to my mind as I was reading the paper, and that I thought of raising in this review, were addressed later on.

The paper thus contributes to advances in the field of distributed sensors in a number of separate ways with a several new concepts and related analysis as well as practical demonstrations.

I think that the separation of the material between the main article and the supplementary information detracts from the readability of the work as a whole; however, I suspect that this separation is dictated by the journal’s policies on length and structure.

I thought that the noise analysis in the supplement was particularly interesting (especially the BOTDA section) and as far as I am aware reveals novel aspects of the properties of noise build-up in pulse-coded BOTDA systems.

Before finalising the manuscript, the authors may want to consider and address the following points.

1. The authors make the statement that time-domain methods are intrinsically best suited for long-distance sensing. I agree with this statement, but I think that the authors should articulate why they believe this to be the case, or reference a definitive source on the subject.

Response: We thank the Reviewer for the suggestion. In the revised manuscript, we have added relevant reasons ‘sensing range is limited by the laser coherence length in frequency-domain approaches, and by background noises originating from uninterrogated fibre positions in correlation-domain approaches’, with associated references.

2. The authors mention the issue of the distortion of the probe pulse sequence caused by insufficient energy storage (as population inversion) in the booster amplifier (EDFA 1 in Fig. 5). The deconvolution approach solves this problem but that is at the expense of a reduction of the peak power for pulses transmitted later in the sequence and, therefore a reduction in the total energy that is launched under optimised conditions. The authors do not discuss the option of pre-emphasising the code sequence to flatten the pulse-height distribution, or indeed of designing an amplifier with sufficient pump power to avoid, or at least alleviate this effect. In Ref [53] (by a subset of the authors of the article under review), the problem was alleviated by amplifying the light between the source and the EOM, so the reasons for moving away from this approach should ideally be explained. It may come down simply to a matter of experimental convenience, but in my view this point should be discussed.

Response: We thank the Reviewer for the comments. We fully agree with the Reviewer that the adaptability to non-uniform sequence is for the ultimate experimental convenience (simplicity), since the experimental layout can be strictly the same as that for standard single-pulse scheme. We were aware that other techniques such as pre-distortion or using additional device can alleviate the nonuniformity of the coding sequence; however, it practically remains extremely difficult to make the sequence perfectly uniform (see Section 5 in [53]). This means that it is eventually inevitable that the coded sequence shows an imperfect uniformity. It is right that Simplex coding shows a reduced sensitivity to this issue, that turns out to be very harmful to Golay codes in which even a slight non-uniformity leads to decoding distortions (see Section 5 in [53]). In this sense the adaptability provided by the proposed code is definitely advantageous.

We have added relevant statements in Discussion (in the 4th item describing Table 2).

3. The authors state a receiver bandwidth of 125 MHz (and sampling rate of 250 MSPS); this bandwidth is well in excess of that required for even 1 m spatial resolution, let alone the 2 m resolution used in some experiments. Did the authors apply any filtering prior to acquisition or in the digital domain to their data?

Response: We thank the Reviewer for raising this question. We did perform filtering in the digital domain (i.e., digitally reducing the noise bandwidth to match the signal bandwidth) for all experimental data measured by both single-pulse and coded schemes. We have actually mentioned this point implicitly in the last sentence of Introduction that ‘It must be emphasized that all demonstrations ... minimised noise bandwidth³¹ ...’, in which the minimum noise bandwidth means the signal bandwidth. To make this point clearer, in the revised manuscript we have added a relevant description in the same sentence, and emphasized this point again in the Discussion (in the paragraph right below newly added Table 2).

4. regarding Fig. 5, I could not see where the heated zone is located along the fibre; is it at the same location as Fig. 4 (d)? Actually, there are two Fig. 5 in the main paper – there appears to be a numbering error.

Response: We thank the Reviewer for pointing out this confusing figure. The original Fig. 5 only shows the retrieved temperature evolution at the hotspot, but did not show the hotspot location. To make results more intuitive, we have newly added a 2D map of retrieved temperature as function of fibre position and acquisition time as Fig. 5(a), in which a 2-m hotspot located at 10.176 km can be clearly seen. In addition, we separate original Fig. 5 and its inset to be Fig. 5(b) and (c), for more clarity.

We have also corrected the numbering error for the figures, thank you for raising our attention to this negligence.

Once, again, I believe that this manuscript is a valuable contribution to the field and my recommendation is that it should be published, ideally with some comments from the authors on the points above.

Yours Sincerely,

Zhisheng Yang,

Luc Thévenaz

(On behalf all authors)

REVIEWER COMMENTS

Reviewer #1 (Remarks to the Author):

The authors addressed the concerns of my previous review in a satisfactory manner. I recommend that the revised manuscript is accepted for publication.

Reviewer #2 (Remarks to the Author):

The authors have satisfactorily implemented the suggested changes and the manuscripts seems now suitable for publication with Nature Communications.

Reviewer #3 (Remarks to the Author):

Feedback on response to the general comments and comment item 1:

We thank the authors for the time they dedicated to the response to our general comment on lack of novelty and further clarifications on our concerns. The authors went into a detailed comparison and contrast of their method with other coding schemes.

However the point was not to make a comparison with other coding schemes but justify the novelty of their contribution.

We also think that the authors made some inaccurate comments on other coding techniques when they reply to our general comment saying that coding to improve SNR is not new, which they also agree with.

In the following, we explain specifically why we do not fully agree with the authors' comments against other coding schemes.

The authors are requested to correct some details on the newly added content in the manuscript, which we believe are inaccurate and unjustifiable as they stand now.

First, while appreciating the authors' genuine efforts to further highlight their contribution, it is worth noting that it is implicitly understood by the fiber optic sensing community that achievable theoretical coding gain refers to gain in the presence of individual noise contributions from response of pulses that are uncorrelated, zero mean random variables with a certain variance, and theoretical values are obtained assuming receiver linearity.

In other words, the authors are simply saying that coding gain with cyclic coding, which is known to be effective in certain conditions, is not effective in specific, "optimized pulse" conditions where it is obviously not expected to yield gain anyways.

It is well known that the coding gain in distributed optical fiber sensor systems can be exploited not only to maximize the sensing distance but also to reduce the required pulse peak power and the measurement times making the measurement effective for many applications.

In practical distributed optical fiber sensing schemes the pulse peak power is not always chosen to be the maximum. In order to work closed to the modulation instability threshold, one would require expensive lasers and amplification schemes making the sensor system not practical for real applications.

Also in some applications, the measurement time is a key factor and in such cases the coding gain can be exploited to reduce it.

This is specified only to say that the conclusion of the authors is sensible but not relevant in many practical schemes.

Even then, the conclusion that "All results indicate that there is no benefit brought by Cyclic coding when comparing with a fully optimized single-pulse scheme", besides not being a strong claim as it precludes an entire range of "non-optimized", but still relevant, cases, is in itself inaccurate. It can not be made for all sensing distances and all sensing schemes, and not for all parameters of measurement as explained in more detail below.

First, even going by the authors' reasoning and data, in Figs. r1 (b) and (c) the noise for cyclic coding is significantly lower than that of the aperiodic one for the first 20 km, which is a measurement range large

enough to be suitable for many areas of applications. To the contrary, the noise for the aperiodic code in this range is more than even the single pulse one. It can also be clearly seen in Figs. r2 (a) and (b), that cyclic coding offers significantly better SNR in the first 20 km than the single pulse while the aperiodic code can not.

Indeed, this is also evident from the reference they cited where it is explicitly stated "...beyond a certain dynamic range...".

In addition, in reference to Fig. r2, it must be noted that averaging a signal whose SNR is already improved with coding will not improve the SNR beyond a threshold since it is the same type of noise [constituted of uncorrelated, zero mean random variables with a certain variance.] which is being targeted by coding and averaging.

Besides, please consider the fact that SNR improvement with a given cyclic code can be obtained in one trace cycle only, while when using a single pulse with averaging, the measurement needs to be made repeatedly. For instance in Raman and BOTDA schemes, this means 10s or hundreds of thousands of acquisitions, while significantly smaller number of averages can be used with cyclic coding effectively reducing the measurement time. This is not to mention the associated data storage, which scales with distance. This has been the key reasons for proposing cyclic coding compared to single pulse and other coding schemes.

Regarding the claim on decoding time, it is not clear how the authors arrived at the assertion starting at line 323:

"a negligible post-processing (decoding) time, enabling real-time on-line fast sensing. This is advantageous over all other conventional codes, especially Cyclic and Simplex codes, as summarized in the third row of Table 2. For instance, in the case of a 100 km-long sensing range using a sampling rate of 250 MS/s and a 255-bit code (i.e. $Nh = 250000$, $Nu = 255$), the evaluated decoding time of Cyclic and Simplex codes is 1806 times longer than that of the proposed GO-code."

As per the statement in the manuscript in line 132: " In addition, the decoding process can be made very fast since the DFT can be computed via fast Fourier transform (FFT)", the faster decoding time using their method is also owed to the implementation of the IDFT computations using FFT.

Are the authors comparing a computation made with FFT for GO-Code with that made without it for the cyclic one? Note that direct matrix multiplication have a complexity which scales with the square of the sample size N order $O(N^2)$, while that of FFT scales with order $O(N \log N)$, which is also reflected in the time complexity of their method stated in table 2, which must be revised appropriately.

How is it conceivable that an operation of decoding to have such a significant difference with one which requires IDFT if the two are obtained using a similar method, and computations are made on the same platform which is not optimized in favor of FFT, which is the case for many processors ? (see for instance:

https://www.alcf.anl.gov/files/slides%20jeongnim%20kim%20FFT_MKL_Applications.pdf)

Please consider that matrix multiplication can also be done using FFT, please see:

<https://www.math.brown.edu/~res/MathNotes/FFT.pdf>

Namely, the claims of faster decoding enabled also by FFT cannot be exclusively attributed to their method and it cannot be precluded from other coding techniques. The authors must either provide additional evidence or withdraw this specific claim and also the subsequent assertion in Table 2.

We also find a problem in the author's assertion that other coding schemes are not robust when there are amplitude fluctuations. As indicated in the basic theory by M. D. Jones, "Using simplex codes to improve OTDR sensitivity," in IEEE Photonics Technology Letters, vol. 5, no. 7, pp. 822-824, this is not true since variations in pulse amplitudes do not always result in measurement artifacts.

In summary, the authors' assertion cannot consider even what their own experiment partly shows and fails to take into account the higher measurement speed merits of cyclic coding and the fact that a method not exclusive to their scheme (FFT) is the reason for the faster measurement time. We think they are drawing conclusions which are inaccurate and unjustifiable.

It must be stressed that the authors should withdraw the misleading claims from the text of the manuscript

and Table 2, or even modification of it if it fails to specify the dynamic range threshold, neglects the benefit of the measurement speed in cyclic coding and the fact that FFT can bring similar benefits in other coding schemes as well, unless they provide additional data.

Feedback on response to comment item 2: we thank the authors for this specific explanation but kindly note that the response does not refer to the intended request for clarification. To elaborate further, what is referred to in the original feedback is the error introduced because of the inevitable possibility of changes (drifts) in this function in a relatively long measurement session. The authors are implicitly assuming that a single calibrated value can be repeatedly used for subsequent measurements within a measurement session which can be long, and at least this assumption should have been explicitly stated in the manuscript.

Feedback on response to comment 3: we thank the authors for the clarification and modification to text, which are sufficient.

Feedback on response to comment 4: we thank the authors for the response and clarification, which are sufficient.

Feedback on response to comment 5: please note that the limitations which were referred to here are specifically the ones in terms of computational complexity of the algorithm, which is apparent in the pseudocode in line 499 as the computation needs to go through two loops, until desired convergence is achieved. Not just in their case, but genetic algorithms are naturally known to have this feature. In fact, due to this inherent complexity, genetic algorithms are one of the computational models used to evaluate and compare the computational performance of potentially high performance processors, and studying complexity of genetic algorithms is a field on its own: please see:

https://backend.orbit.dtu.dk/ws/portalfiles/portal/127386958/TCS_D_13_00759R2.pdf

Hence, what remains is explaining the impact of this computational complexity on the time of convergence in obtaining the best coded sequence. As long as the authors think more powerful tools and methods result in improvements, this issue needs to be accounted for because they are saying their method allows the use of arbitrary code length and the best sequence at each codeword length is determined using this algorithm.

In conclusion, we reiterate our humble opinion that the paper is of high technical level but lacks of sufficient novelty for publication on Nature Communications.

Reviewer #4 (Remarks to the Author):

This review refers to the revised version (242135-1).

I was already quite happy with the original version and the questions and suggestions that I raised after V0 have been fully addressed.

Out of personal interest, I also looked rather briefly at the comments by other reviewers and the response by the authors and by and large they have also been answered in full (.).

Overall, the paper has improved in clarity, a couple of errors have been removed and the new structure of the paper (which I disliked in the original submission) makes it much more readable now, in my view.

My recommendation to the Editor to accept the paper as it now stands.

We sincerely appreciate the time you dedicated to evaluate our response letter in the first round. While Reviewers #1, #2 and #4 have approved the novelty and quality of our work, we have thoroughly considered all comments from Reviewer #3 and revised our manuscript every time it is applicable. Our detailed responses to your specific questions are listed below.

Reviewer 3:

Feedback on response to the general comments and comment item 1:

We thank the authors for the time they dedicated to the response to our general comment on lack of novelty and further clarifications on our concerns. The authors went into a detailed comparison and contrast of their method with other coding schemes.

However the point was not to make a comparison with other coding schemes but justify the novelty of their contribution.

Response: We are confused by the comment that ‘however the point was not to make a comparison’. The comparison was suggested by the Reviewer, who raised the concern that ‘other approaches can achieve similar performance to our proposed technique, which makes our work lack of novelty’. It would be an impossible task to prove that our technique outperforms other existing methods without performing a comparison, being an evidence justifying the novelty.

The Reviewer will certainly agree with us that “novelty” is a concept that may be tainted by subjectivity. Although we highly respect the personal opinion of the Reviewer, we estimate, like the other 3 Reviewers, that our technique clearly brings all flavours of novelty, since 1) this type of coding has never been proposed before and is conceptually new, 2) it is an objective scientific and technical progress since it offers better performance than reported and proved configurations, without impacting the experimental complexity. We have made all efforts described below to avoid being placed in a trivial conflict of subjective opinions with our responses.

We also think that the authors made some inaccurate comments on other coding techniques when they reply to our general comment saying that coding to improve SNR is not new, which they also agree with.

Response: In the response letter of the first-round reviewing, we only agreed with the fact that ‘coding to improve SNR is not new’; this is of course not new, since it is the common target shared by all coding techniques to improve the performance of a distributed sensor, which is ultimately limited by SNR for all specifications.

We must emphasise again that this cannot lead to the conclusion that our work is not novel, otherwise all coding techniques were not novel at the time people proposed them, except the very first one which was at the era of telegraphs. The added value of our manuscript is the novel concept behind our coding, which is not transposed from applications in other fields, in contrast with other former reports on coding in fibre sensing.

In the following, we explain specifically why we do not fully agree with the authors’ comments against other coding schemes.

The authors are requested to correct some details on the newly added content in the manuscript, which we believe are inaccurate and unjustifiable as they stand now.

First, while appreciating the authors’ genuine efforts to further highlight their contribution, it is worth noting that it is implicitly understood by the fiber optic sensing community that achievable theoretical coding gain refers to gain in the presence of individual noise contributions from response of pulses that are uncorrelated, zero mean random variables with a certain variance, and theoretical values are obtained assuming receiver linearity.

Response: We thank the Reviewer for highlighting that ‘the impact of noise on coding was implicitly understood by the fiber optic sensing community’. This justifies why we provided Supplementary notes 3 and 4 to **explicitly** explain

the theory. This is definitely new, rigorously demonstrating the contribution of our work to the community as well as the novelty, as it has been also acknowledged by the other reviewers.

In other words, the authors are simply saying that coding gain with cyclic coding, which is known to be effective in certain conditions, is not effective in specific, “optimized pulse” conditions where it is obviously not expected to yield gain anyways.

It is well known that the coding gain in distributed optical fiber sensor systems can be exploited not only to maximize the sensing distance but also to reduce the required pulse peak power and the measurement times making the measurement effective for many applications.

In practical distributed optical fiber sensing schemes the pulse peak power is not always chosen to be the maximum. In order to work closed to the modulation instability threshold, one would require expensive lasers and amplification schemes making the sensor system not practical for real applications.

Response: In our manuscript, we only addressed ‘cyclic coding brings no benefit to optimised single-pulse scheme’, but we never claimed and even intended to claim ‘cyclic coding is useless in all conditions’! This is a misinterpretation of what we intend to mean.

Moreover, while agreeing with the Reviewer that ‘cyclic coding can be effective in non-optimised conditions’, we do not agree with the Reviewer to define ‘optimised pulse’ as ‘specific condition’. Instead, here we emphasise that ‘the optimised pulse’ is a very common and dominant condition in both academia and industry; the ‘non-optimised pulse’ is actually a specific and rare condition, for niche applications not substantiated by a noticeable marketshare today.

We also do not share the Reviewer’s opinion that ‘using amplification schemes to optimise the pulse power is expensive and not practical for real applications’. Even though this view may be widespread in the community and belongs to the common knowledge in the early years of coding in distributed fibre sensing, it has been seriously revisited by recent works, in which the authors substantially, but not uniquely, contributed. These works sometimes highly perturbed our convictions and this needed to deeply revisit our view about coding: in many situations coding brings a meaningless benefit with respect to an optimised configuration.

We fully understand the point of view of the Reviewer, but it is substantiated by neither any practical evidence, nor an existing experimental demonstrator publicly released. **The common view that the EDFA is an expensive element is actually outdated and totally challenged by the reality: it contributes to a few percents (less than 5%) to the cost of a BOTDA system, much less than the computing hardware which is the most costly element. Thus, it turns out to be economically more favourable to implement an optimised configuration than adding hardware and computing complexity to implement state-of-the-art coding.** This conclusion does not belong to our personal speculation, but results from public and personal communication from the CTO of an established company with a global leading position in Brillouin distributed sensors.

To objectively make this point clear, we have modified the relevant sentence in last paragraph of Supplementary note 3, as ‘All results indicate that there is no benefit brought by Cyclic coding when comparing with a fully optimised single-pulse scheme for any sensing range. Note that, for a non-optimised pulse power, cyclic coding can be effective as demonstrated by previous publications, though this non-optimised condition has not yet proved to bring any global advantage in real conditions.’

We are definitely ready to revisit our convictions, which are based on the latest theoretical studies confirmed by experimental demonstrations, if it can be demonstrated that a better system can be realised (economically and in term of performance) using a non-optimised configuration exploiting classical or cyclic coding. But as today it is purely speculative and has not yet been demonstrated, so in full rigour this situation cannot be used to deny the impact of our work, to our humble opinion.

Also in some applications, the measurement time is a key factor and in such cases the coding gain can be exploited to reduce it.

This is specified only to say that the conclusion of the authors is sensible but not relevant in many practical schemes.

Even then, the conclusion that “All results indicate that there is no benefit brought by Cyclic coding when comparing with a fully optimized single-pulse scheme”, besides not being a strong claim as it precludes an entire range of “non-optimized”, but still relevant, cases, is in itself inaccurate. It can not be made for all sensing distances and all sensing schemes, and not for all parameters of measurement as explained in more detail below.

First, even going by the authors’ reasoning and data, in Figs. r1 (b) and (c) the noise for cyclic coding is significantly lower than that of the aperiodic one for the first 20 km, which is a measurement range large enough to be suitable for many areas of applications. To the contrary, the noise for the aperiodic code in this range is more than even the single pulse one. It can also be clearly seen in Figs. r2 (a) and (b), that cyclic coding offers significantly better SNR in the first 20 km than the single pulse while the aperiodic code can not.

Indeed, this is also evident from the reference they cited where it is explicitly stated “...beyond a certain dynamic range...”.

Response: First of all we would like to draw the Reviewer’s attention that, even in the first version of our manuscript, we have clearly mentioned that ‘the code gain can be exploited to reduce the measurement time’ in the Discussion section. We do not see why we are uncredited for this evidence.

We must emphasise the other evidence that the measurement time can be reduced only when there is a given coding gain. In the first response letter, we have provided the experimental data showing that there is no actual coding gain provided by cyclic coding for a 100 km range, so it is definitely not possible to use cyclic coding to reduce the measurement time over this long range eagerly asked by the end users of many practical applications.

We must also remind that the **only metric to qualify any distributed sensor is to evaluate the response at the lowest SNR position**, which is usually at the fibre far-end. Here may come the confusion made by the Reviewer: if a system based on a given configuration may provide a better response over the first kilometres than our aperiodic coding, it cannot be concluded that it is better for a shorter distance range. Only the response at the lowest SNR position matters for a given distance range. The evaluation must be carried out from scratch if the distance range is shorter, since the pulse and probe signals are conditioned very differently. So, it is meaningless that cyclic coding can provide higher SNR than aperiodic code only in first 20 km, where the SNR with cyclic coding is identical to the single-pulse scheme. This does not change the fact that the cyclic coding brings no benefit to optimised single-pulse scheme.

So, we stress again on the essential fact that the response for a shorter range cannot be simply interpolated from the response on the short initial segment from a longer-range measurement. Actually, even for applications over 20 km as proposed by the Reviewer, the cyclic coding does not provide any SNR improvement as well. This is clearly demonstrated by newly obtained experimental results, using a 25 km long sensing fibre (we currently don’t have a fibre of exactly 20 km length). Fig. r1(a) and (b) here below show the noise profile of aperiodic code and cyclic code, respectively, while Fig. r1(c) shows the SNR obtained over the entire sensing fibre for both aperiodic and cyclic codes as well as single pulse (measured with the same number of averaged and same measurement time). It can be clearly found that both codes cannot provide decisive SNR improvement along a 25 km range (in an actual 25 km-long sensing fibre), due to the additional signal-dependent noise induced by the code sequences themselves (noise that dominates the measurement). However, it is very relevant to notice that the total noise in the case of cyclic code remains constant all over the measured trace (see Fig. r1(b)), while the noise reduces with distance for the aperiodic code (see Fig. r1(a)). This behaviour is essentially due to the pulse distribution over the sensing fibre, explaining the higher SNR improvement provided by the proposed GO-code at longer sensing ranges compared to cyclic coding.

Fig. r1. Noise profiles for (a) aperiodic code and (b) cyclic code. (c) the SNR profile of single-pulse, aperiodic code and cyclic code

In short, we do not agree with the Reviewer's comment that 'the original statement is inaccurate since cyclic coding may be effective at other sensing range'. Here, supported by the standard metric, the derived expressions in Supplementary note 3 and the experimental results, we demonstrate that cyclic coding brings no benefit to optimised single-pulse scheme for any sensing range. Therefore, our original statement that 'cyclic coding brings no benefit to optimised single-pulse scheme' holds true and is experimentally evidenced, to the best of our recent knowledge. We believe that this demonstration, based on a rigorous scientific analysis, is indeed an important and novel contribution of our manuscript, correcting and updating the current scientific knowledge on cyclic codes.

In addition, in reference to Fig. r2, it must be noted that averaging a signal whose SNR is already improved with coding will not improve the SNR beyond a threshold since it is the same type of noise [constituted of uncorrelated, zero mean random variables with a certain variance.] which is being targeted by coding and averaging.

Response: We do not share the Reviewer's opinion on this point. Unless there is a limitation by the resolution in the analog-to-digital conversion, which is a technical limit that can be corrected, there is no identified fundamental reason why averaging a signal cannot further improve the SNR that has been already improved by coding. It is unquestionable to our opinion that there is no sort of fundamental threshold as mentioned by the Reviewer. It is always possible to condition the analog signal and to improve the ADC conversion, so that no such threshold is limiting in a given practical configuration.

There is no fundamental limit in reducing the noise by averaging if noise keeps uncorrelated. We kindly ask the Reviewer to argue in depth on this point if he maintains his claim.

Besides, please consider the fact that SNR improvement with a given cyclic code can be obtained in one trace cycle only, while when using a single pulse with averaging, the measurement needs to be made repeatedly. For instance in Raman and BOTDA schemes, this means 10s or hundreds of thousands of acquisitions, while significantly smaller number of averages can be used with cyclic coding effectively reducing the measurement time. This is not to mention the associated data storage, which scales with distance. This has been the key reasons for proposing cyclic coding compared to single pulse and other coding schemes.

Response: As already discussed in previous responses, when there is no coding gain (due to the additional signal-dependent noise induced by the code itself), there is no measurement time improvement. This is the case of cyclic coding, with optimised power for any sensing range, as experimentally demonstrated in the previous response and the figures in Supplementary note 3. Fig. r1(c) also shows that cyclic code results in the same SNR as the single pulse case over a 25 km-long fibre when using the same number of averaged traces, demonstrating no reduction in the measurement time for this fibre sensing range.

Therefore, we do not agree with the Reviewer on this point. In other words, cyclic coding cannot bring measurement time improvement when compared to an optimised pulse power scheme, which remains so far the only standard applied in the real world. We hope that our novel aperiodic coding will contribute to break this stop wall and we made all efforts to rigorously validate this breakthrough in our manuscript.

Regarding the claim on decoding time, it is not clear how the authors arrived at the assertion starting at line 323:

“a negligible post-processing (decoding) time, enabling real-time on-line fast sensing. This is advantageous over all other conventional codes, especially Cyclic and Simplex codes, as summarized in the third row of Table 2. For instance, in the case of a 100 km-long sensing range using a sampling rate of 250 MS/s and a 255-bit code (i.e. $Nh = 250000$, $Nu = 255$), the evaluated decoding time of Cyclic and Simplex codes is 1806 times longer than that of the proposed GO-code.”

As per the statement in the manuscript in line 132: “ In addition, the decoding process can be made very fast since the DFT can be computed via fast Fourier transform (FFT)”, the faster decoding time using their method is also owed to the implementation of the IDFT computations using FFT.

Are the authors comparing a computation made with FFT for GO-Code with that made without it for the cyclic one? Note that direct matrix multiplication have a complexity which scales with the square of the sample size N order $O(N^2)$, while that of FFT scales with order $O(N \log N)$, which is also reflected in the time complexity of their method stated in table 2, which must be revised appropriately.

How is it conceivable that an operation of decoding to have such a significant difference with one which requires IDFT if the two are obtained using a similar method, and computations are made on the same platform which is not optimized in favor of FFT, which is the case for many processors ? (see for instance:

https://www.alcf.anl.gov/files/slides%20jeongnim%20kim%20FFT_MKL_Applications.pdf)

Please consider that matrix multiplication can also be done using FFT, please see:

<https://www.math.brown.edu/~res/MathNotes/FFT.pdf>

Namely, the claims of faster decoding enabled also by FFT cannot be exclusively attributed to their method and it cannot be precluded from other coding techniques. The authors must either provide additional evidence or withdraw this specific claim and also the subsequent assertion in Table 2.

Response: We appreciate the Reviewer for reminding us the possibility to decode cyclic coding in a fast way, this definitely improves the quality of our paper. However, we must correct the Reviewer that the complexity for fast cyclic decoding is actually $2N_h \log_2 N_u$, rather than $2N_u \log_2 N_u$ as proposed by the Reviewer. For the exemplified parameters in the Discussion, the decoding time of cyclic coding can be 3 times faster than the proposed GO-code. However, we would like to draw the Reviewer’s attention that this simply means the decoding times for both codes are negligible compared to the measurement time. In other words, the fast-decoding feature of the propose GO-code still holds. We have modified relevant statement in Discussion section.

Note also that, this does not change the fact that ‘cyclic coding brings no benefit to optimised single-pulse scheme’.

We also find a problem in the author’s assertion that other coding schemes are not robust when there are amplitude fluctuations. As indicated in the basic theory by M. D. Jones, "Using simplex codes to improve OTDR sensitivity," in IEEE Photonics Technology Letters, vol. 5, no. 7, pp. 822-824, this is not true since variations in pulse amplitudes do not always result in measurement artifacts.

Response: We would like to clarify that we never claimed ‘Simplex coding is not robust to amplitude fluctuations’, instead, we only claimed ‘simplex coding is not robust to amplitude non-uniformity’. This is a major and fundamental nuance of high relevance here.

We would like to draw the Reviewer’s attention that the ‘amplitude non-uniformity’ also includes ‘decaying amplitude’, and it has been addressed by many times in our manuscript.

The unquestioned fact is that Simplex coding can be robust to amplitude fluctuations, but is not robust against decaying amplitude (e.g. resulting from EDFA amplification), as has been clearly demonstrated in ref 53 of the manuscript.

In summary, the authors' assertion cannot consider even what their own experiment partly shows and fails to take into account the higher measurement speed merits of cyclic coding and the fact that a method not exclusive to their scheme (FFT) is the reason for the faster measurement time. We think they are drawing conclusions which are inaccurate and unjustifiable.

It must be stressed that the authors should withdraw the misleading claims from the text of the manuscript and Table 2, or even modification of it if it fails to specify the dynamic range threshold, neglects the benefit of the measurement speed in cyclic coding and the fact that FFT can bring similar benefits in other coding schemes as well, unless they provide additional data.

Response: As a brief summary of all our responses, we clearly emphasise that, once the fact of the absence of benefit from classical and cyclic coding is acknowledged and accepted, which has been a long term revisiting even by the authors to convince themselves, most of Reviewer's comments turn inapplicable to demonstrate the inanity of our novel approach, and we only agree with one comment that 'the decoding time (not the measurement time) of cyclic code can be faster than the proposed GO-code'.

We sincerely appreciate this correct comment from the Reviewer, which can definitely improve the scientific rigour of our paper. We collegially hope that our explanations backed by experimental demonstrations will convince the Reviewer that the common sense about coding has to be deeply revisited when applied to distributed fibre sensing.

Feedback on response to comment item 2: we thank the authors for this specific explanation but kindly note that the response does not refer to the intended request for clarification. To elaborate further, what is referred to in the original feedback is the error introduced because of the inevitable possibility of changes (drifts) in this function in a relatively long measurement session. The authors are implicitly assuming that a single calibrated value can be repeatedly used for subsequent measurements within a measurement session which can be long, and at least this assumption should have been explicitly stated in the manuscript.

Response: We still do not know what is 'inevitable possibility of changes in the function' meant by the Reviewer. We still insist that we do not have any drift during experiments over several months, since our system is designed accordingly based on our decades-long experience. We are glad to confirm that this issue does not belong to our reality.

Nevertheless, we are now aware thanks to the Reviewer that it may be not the case everywhere. Therefore, we have added a sentence in line 124, as 'Such calibration can be made for a long measurement session, provided that the experimental stability is sufficient and there is no uncontrolled signal drift of significant importance in the instrumentation.'

Feedback on response to comment 3: we thank the authors for the clarification and modification to text, which are sufficient.

Feedback on response to comment 4: we thank the authors for the response and clarification, which are sufficient.

Feedback on response to comment 5: please note that the limitations which were referred to here are specifically the ones in terms of computational complexity of the algorithm, which is apparent in the pseudocode in line 499 as the computation needs to go through two loops, until desired convergence is achieved. Not just in their case, but genetic algorithms are naturally known to have this feature.

In fact, due to this inherent complexity, genetic algorithms are one of the computational models used to evaluate and compare the computational performance of potentially high performance processors, and studying complexity of genetic algorithms is a field on its own: please see:

https://backend.orbit.dtu.dk/ws/portalfiles/portal/127386958/TCS_D_13_00759R2.pdf

Hence, what remains is explaining the impact of this computational complexity on the time of convergence in obtaining the best coded sequence. As long as the authors think more powerful tools and methods result in improvements, this issue needs to be accounted for because they are saying their method allows the use of arbitrary code length and the best sequence at each codeword length is determined using this algorithm.

Response: We feel important at this stage to remind that the complexity in the ‘searching process’ does not compromise any performance when using the code for distributed sensors, as the codes are computed once forever. The logic is simply similar to machine learning, in which the ‘training process’ can be very complex and may take a fairly long time, however, the target is to make a fast ‘testing process’.

The use of our algorithm is actually even a smart way to fully exploit the computational resources: we allocate more complexity in the searching process, to achieve a top performance while using the delivered code.

So, again, we cannot follow the Reviewer’s opinion that ‘the complexity of the searching process is a problem’.

In addition, we would like to draw the Reviewer’s attention that, when we said ‘arbitrary code length’, the context means ‘arbitrary length required by a given system’, with corresponding F_E not larger than 200 as has already been addressed in the first response letter. To make this point clear, we have modified the relevant sentence as ‘Arbitrary F_E required by a given system’ in Table 2.

In conclusion, we reiterate our humble opinion that the paper is of high technical level but lacks of sufficient novelty for publication on Nature Communications.

Response: As mentioned in our preamble the concept of novelty can be undermined by subjectivity and we respect the Reviewer’s appreciation. We substantiated our claim of novelty based on the latest knowledge in coding for fibre sensing, by proposing a novel code - un-inspired from previous works in different fields unlike Simplex, Golay and cycling coding - that removes a serious deadlock in the implementation of coding in sensing. We are ready to revisit our convictions if it is objectively contradicted by up-to-date knowledge and experiments.

The fact remains that the proposed GO-code has never been proposed before, and the overall performance of the proposed GO-code is superior to other existing coding scheme (definitely including cyclic coding), which we believe is clearly demonstrated in this revised version of the paper.

If the Reviewer still doubt on the novelty and the relevance, we would appreciate to be contradicted and challenged by kindly and respectfully asking the Reviewer to provide the proof of a coding technique that can achieve similar performance as the proposed GO-code, for instance: same setup as optimised single-pulse BOTDA scheme, 2 m spatial resolution, 100 km sensing range, 1024 averages, 0.63 MHz BFS uncertainty, 2.8 min real measurement time (the same as single-pulse scheme), 1.8 s decoding time.

[redacted]

Yours Sincerely,

Zhisheng Yang,

Marcelo A. Soto

Luc Thévenaz

(On behalf of all authors)

REVIEWERS' COMMENTS

Reviewer #4 (Remarks to the Author):

In previous reviews and in particular after the first revision, I had recommended publication of this paper. However, as requested by the Editor, I have now read the further comments by Reviewer 3 and the response by the authors.

I have the following comments.

There are several quite new aspects to this paper, in particular

a) the demonstration of aperiodic codes of arbitrary length, whereas previously known codes, e.g. Golay or Simplex, were limited to lengths measured as powers of 2 or (powers of 2)-1. This allows a finer optimisation of the code. Frankly, I am not sure how significant a completely free choice of the code length is compared with picking the nearest power of 2, but at least the authors have demonstrated that they can reach an optimum code length free of previously held constraints.

b) the demonstration of a decoding process based on deconvolution, rather than correlation, which I believe makes the coding more robust to imperfections such as non-linearities in the generation of the optical version of the pulse code and (as discussed in the paper), the decay due to depletion of the population inversion in an optical amplifier. It should be noted that deconvolution has been used in previous work for correcting coding imperfections [G. D. B. Vazquez, O. E. Martínez, and D. Kunik, "Distributed Temperature Sensing Using Cyclic Pseudorandom Sequences," *IEEE Sens J*, vol. 17, no. 6, pp. 1686-1691, 2017.] but not as far as I am aware as the primary method for decoding in pulse compression.

c) the optimisation of the process for selecting the code length using the GO approach.

d) the supporting demonstrations and noise theory (particularly in the Supplementary notes) are also, in my opinion, new to the literature

To my mind, the points above amply justify the paper being published and I regret that its appearance has been delayed by the debate between Reviewer 3 and the authors.

The points above speak directly against the first objection by Reviewer 3, on the insufficiency of novelty in this work.

To clarify the issue of novelty of coding schemes, clearly they have been known in the field of OTDR since the 1980s and actually also in distributed sensing in the same decade [J. K. A. Everard, Greatly enhanced spatial detection of optical backscatter for sensor applications, WO87/07014, 1987.] However, the prior existence of coding schemes in distributed sensing in no way vitiates the comments that I made (a-d above) on what I consider to be clear advances over the known state of the art proposed and demonstrated by the authors in this paper.

The supplementary notes, particularly the simulations of the long-range relative performance of the cyclic coding vs single-pulse coding clarify the need for using finite-length codes. Although the reasons for the result have been interiorised by those of us who have worked in this field for a while, I am not aware of this analysis having been presented before.

Regarding the comment:

"In other words, the authors are simply saying that coding gain with cyclic coding, which is known to be effective in certain conditions, is not effective in specific, "optimized pulse" conditions where it is obviously not expected to yield gain anyways.

It is well known that the coding gain in distributed optical fiber sensor systems can be exploited not only to maximize the sensing distance but also to reduce the required pulse peak power and the measurement times making the measurement effective for many applications.

In practical distributed optical fiber sensing schemes, the pulse peak power is not always chosen to be the maximum. In order to work closed to the modulation instability threshold, one would require expensive lasers and amplification schemes making the sensor system not practical for real applications."

Clearly, it is always possible to design a poor sensor and then show that a particular technique improves the outcome. For example, a distributed sensor with a poorly designed receiver would benefit from cyclic coding under certain conditions; this is why the comparison with an optimised single-pulse condition is relevant to this paper.

I support the authors' comment on the present convenience and cost-effectiveness of optical amplification, at least at wavelengths where doped-fibre amplifiers are widely available. However, under conditions where the optimum launch power (limited by non-linear effects) cannot be used, there may well be a case for

continuous codes.

On the point regarding the value of cyclic coding at short range, I support the authors' argument that the benefit must be quantified at the most distant point for which the system is designed. Additionally, for a shorter range, the system would be re-optimised, for example for pulse power and pulse repetition frequency which means that the comparison between two approaches (e.g. single pulse vs cyclic coding) must be made afresh for each condition.

I agree with the authors that there is nothing to prevent a coding-enhanced system to further benefit from averaging. In fact, this is usually assumption going back to the fundamental paper on complementary compression coding [42].

I think that the arguments about decoding time become irrelevant if a) the decoding time is sufficiently small relative to the acquisition and averaging time and b) provided that the hardware involved in keeping up with the data update rate is not prohibitive. In systems where considerable averaging is employed, the acquisition time is likely to dominate.

In summary, I feel that the authors have dealt fully with the critique of Reviewer 3 and my personal opinion is that the paper should be published. Whatever further doubts might remain with Reviewer 3 on the merits of the present work compared with cyclic coding, I think that the significant novel material in this manuscript fully justify its publication. There probably are cases where cyclic coding is preferred on engineering grounds (such as the required operating wavelength, availability of sources, etc.); however, the value of cyclic coding in some cases does not detract from the value of the work described in the present manuscript.